# DreamActor-M2: Unleashing Pre-trained Video Models for Universal Character Image Animation via In-Context Fine-tuning

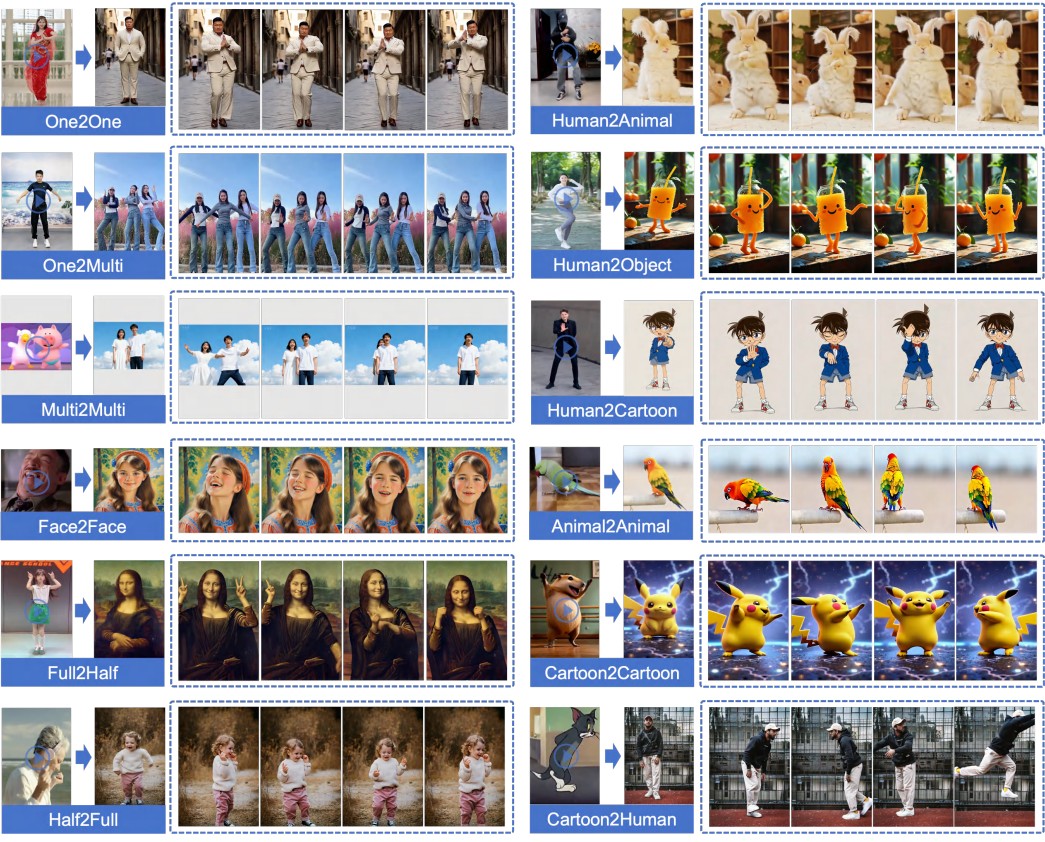

Figure 1: The proposed **DreamActor-M2** exhibits strong generalization capability, producing diverse animations while preserving consistent character appearance. The arrow → denotes the transfer of character motion from the driving video to the character depicted in the reference image.

## ABSTRACT

Character image animation aims to generate high-fidelity videos from a reference image and a driving video, with broad applications in digital humans. Despite recent advances, current methods suffer from two key limitations: reliance on auxiliary pose encoders introduces modality gaps that weaken alignment pre-trained generative priors, and dependence on explicit pose signals severely limits generalization beyond human-centric scenarios. We propose **DreamActor-M2**, a universal framework that redefines motion conditioning through an in-context LoRA fine-tuning paradigm. By directly concatenating motion signals and reference images into a unified input, our approach preserves the backbone's native modality and fully exploits pre-trained capabilities without architectural modifications, enabling plug-and-play motion control consistent with the principles of in-context learning. Furthermore, we extend this formulation beyond pose-driven control to an end-to-end framework that conditions directly on raw video frames, trained by a synthesis-driven data generation pipeline. Extensive experiments demonstrate

that DreamActor-M2 achieves state-of-the-art performance with superior fidelity, controllability, and cross-domain generalization, marking a significant step toward more flexible and scalable motion-driven video generation.

# 1 INTRODUCTION

Character image animation (Tan et al., 2024; 2025; Luo et al., 2025; Li et al.) is an appealing yet challenging task with broad applications in digital entertainment, virtual avatars, film and television production, and human-computer interaction. It aims to generate high-fidelity videos in which the subject's appearance remains consistent with a given reference image, while its motion faithfully follows a driving video. This task imposes stringent constraints: it must concurrently ensure identity preservation (Tu et al., 2025; Chang et al., 2025; Wang et al., 2024b), motion consistency (Ma et al., 2025; Lei et al., 2024), and effective generalization across diverse character types, including humans, cartoons, and animals (Zhang et al., 2025a; Tan et al., 2024; 2025).

Recent advances in large-scale video generation models (Blattmann et al., 2023; Yang et al., 2024; Wan et al., 2025; Kong et al., 2024; Seawead et al., 2025; Gao et al., 2025; Zhang et al., 2025b) have provided powerful pre-trained backbones with remarkable generative capabilities. Building on these foundational models, most contemporary studies (He et al., 2025; Luo et al., 2025; Zhang et al., 2025a; Ding et al., 2025; Wang et al., 2025) have adapted these backbones for character image animation, achieving promising progress. These approaches demonstrate that pre-trained video generators can act as strong priors, significantly reducing the need for training from scratch and enabling rapid transfer to the downstream task of character image animation.

Despite recent progress, most existing methods still suffer from two fundamental drawbacks. **First**, most methods (MooreThreads, 2024; Zhang et al., 2024; Wang et al., 2025; Luo et al., 2025; Ding et al., 2025) inject motion into pre-trained video generation backbones via auxiliary pose encoders. This design inevitably introduces a *input modality gap*, as pose information is separately encoded and fused with the backbone through feature concatenation or cross-attention. Such a mismatch disrupts the consistency with large-scale pre-training, weakening the generative priors learned during pre-training. As a result, the generated outputs often exhibit reduced fidelity and controllability, particularly in complex or unseen scenarios. **Second**, the predominant reliance on explicit pose control signals (e.g., 2D keypoints or 3D SMPL) make current methods highly dependent on pose estimators (Karras et al., 2023; Xu et al., 2024; Wang et al., 2024a; Zhu et al., 2024; Zhang et al., 2024; Peng et al., 2024; Gan et al., 2025; Ding et al., 2025). However, these estimators are error-prone in human-centric videos and fundamentally lack generalizability to non-human domains such as cartoons or animals, severely restricting adaptability. Although implicit motion representations have been explored (Tan et al., 2024; 2025; Song et al., 2025), they still rely on pose-derived supervision and thus remain constrained by the inherent weaknesses of pose estimation.

To address these challenges, we propose **DreamActor-M2**, a universal framework featuring an **In-Context** LoRA fine-tuning strategy that supports both pose- and video-driven control. Unlike prior methods that rely on auxiliary pose encoders, our approach directly concatenates motion control signals with the reference image at the spatial level. This design allows the pre-trained backbone to naturally interpret motion as input context. As a result, the LoRA adapter can align motion and appearance seamlessly without changing the input modality or backbone architecture, achieving a seamless plug-and-play motion injection consistent with the principles of in-context learning.

Our framework is implemented in two powerful variants. The first, **Pose-based DreamActor-M2**, leverages 2D pose skeletons as motion signals to drive character animation. To expand applicability beyond pose-driven settings, we then create **End-to-End DreamActor-M2**, which directly accepts raw video frames as motion conditions. A key obstacle, however, lies in the scarcity of paired data that simultaneously ensures motion fidelity and cross-identity diversity. To solve this, we devise a novel two-stage synthesis-and-training strategy. In the first stage, we use pre-trained *Pose-based DreamActor-M2* to automatically generate motion-consistent videos with varied identities by transferring motion from source videos to different reference images. Then, in the second stage, we pair these synthesized videos with the originals to create a large-scale, high-quality pseudo-paired dataset. This enables effective end-to-end training without the need for manual, real-world annotations. By removing the dependency on explicit pose signals, our end-to-end variant can generalize

effortlessly to both human and non-human motion videos, marking an important step toward more flexible and broadly applicable motion-driven generation.

For a comprehensive evaluation, we construct an Evaluation Benchmark covering a wide range of motion categories and character identities. Extensive experiments demonstrate that DreamActor-M2 consistently outperforms prior methods, achieving superior motion consistency and generalization. Our main contributions are summarized as follows:

## 2 RELATED WORK

**Latent Video Diffusion Models.** Diffusion-based generative models (Blattmann et al., 2023; Yang et al., 2024; Wan et al., 2025; Kong et al., 2024; Seaweed et al., 2025; Gao et al., 2025; Zhang et al., 2025b) have recently achieved remarkable success in video generation. More recently, Wan2.1 (Wan et al., 2025) and Seedance 1.0 (Gao et al., 2025) have emerged as high-performance video foundation model, supporting both text-to-video and image-to-video generation. Given that character image animation is inherently more aligned with the image-to-video generation setting, we adopt Seedance 1.0 as the pre-trained video generation backbone for our proposed DreamActor-M2 framework.

**Pose Guidance in Character Image Animation.** Pose-guided approaches remain the dominant paradigm for character image animation. A series of studies (Hu, 2024; Zhu et al., 2024; Xu et al., 2024; Zhang et al., 2024; Wang et al., 2025; Gan et al., 2025; Chang et al., 2025; He et al., 2025) adopt a *pose-aligned* strategy, which uses 2D skeletons via pose estimation as motion control signals. While this enforces spatial alignment and ensures motion consistency, training under same-identity settings often causes *identity leakage*, where identity cues become entangled with motion features, severely degrading performance in cross-identity scenarios. To alleviate this, *alignment-free* strategies (Tan et al., 2024; 2025; Ding et al., 2025) attempt to decouple pose signals from strict spatial alignment. For example, Animate-X (Tan et al., 2024) introduces random skeleton scaling, and MTVCrafter (Ding et al., 2025) normalizes 3D SMPL joints using statistical templates. However, such designs inevitably distort motion semantics, leading to reduced motion fidelity. In contrast, our method adopts an alignment-free design to mitigate identity leakage and further leverage MLLMs to preserve high-level motion intent and improve overall animation quality.

**In-Context Learning.** In-context learning (ICL) (Brown et al., 2020) enables models to adapt to new tasks without parameter updates by conditioning on task-specific examples within the input context. While ICL has achieved remarkable success in LLMs (Brown et al., 2020; Dong et al., 2022), its application in visual generation remains in its infancy, and its potential remains to be further tapped. In this work, we introduce ICL principles into diffusion-based motion-driven video generation by directly concatenating motion signals with the reference image (humans or non-humans). This design allows the backbone to interpret motion as a part of the input context, enabling seamless plug-and-play motion control without any architectural modification.

## 3 METHOD

In this section, we present DreamActor-M2, a unified framework designed to generate realistic animation videos conditioned on a reference image and driving signals (e.g., pose sequences or video clips). Sec. 3.1 provides an overview of the Diffusion Transformer (DiT) backbone. Sec. 3.2 introduces the in-context motion injection strategy. Sec. 3.3 then elaborates on the Pose-based DreamActor-M2 framework. Finally, Sec. 3.4 details the End-to-End DreamActor-M2 pipeline.

### 3.1 PRELIMINARY

**Latent Diffusion Model.** As shown in Fig. 2, our framework is built upon the Latent Diffusion Model (LDM). A pretrained Variational Autoencoder (VAE) is employed to encode the input images $I$ into a latent representation $z = \xi(I)$. During training, Gaussian noise $\epsilon$ is progressively injected into the latent $z_t$ at different timesteps. The model is optimized with the following objective:

$$\mathcal{L} = \mathbb{E}_{z_t, c, \epsilon, t} \left( \|\epsilon - \epsilon_\theta \left( z_t, c, t \right)\|_2^2 \right) \tag{1}$$

where $\epsilon_\theta$ denotes the denoising network and $c$ represents conditional input. At inference time, noise latents are iteratively denoised and reconstructed into images through VAE's decoder. In this

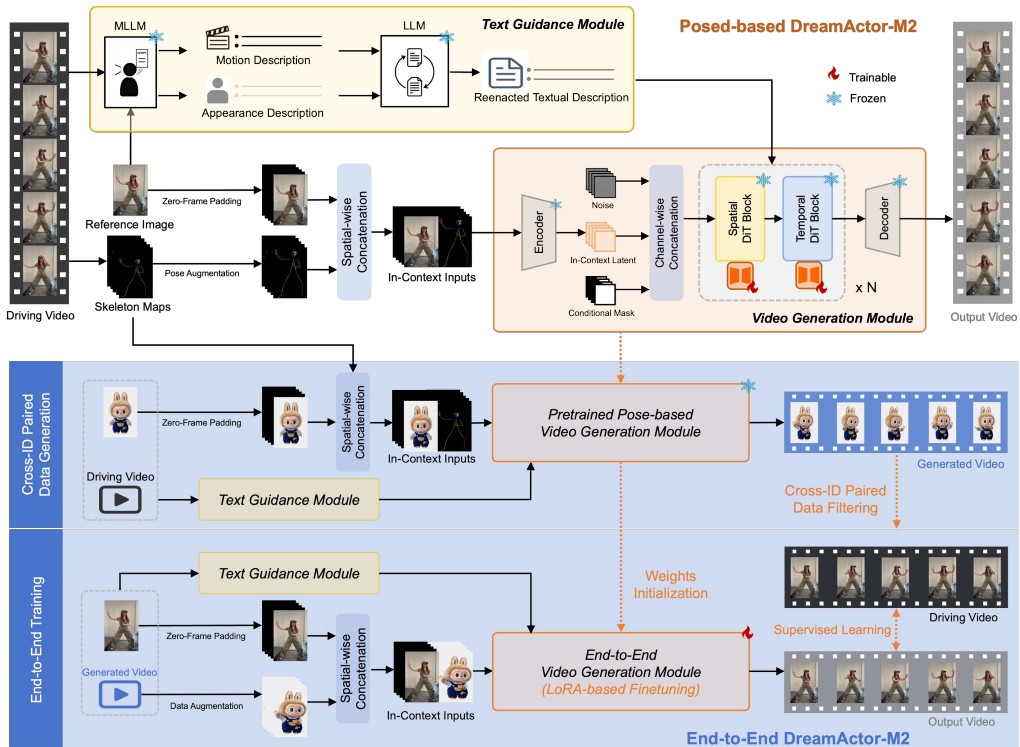

Figure 2: The overview of proposed DreamActor-M2.

work, we adopt Seedance 1.0 (Gao et al., 2025) as the video generative backbone, which follows the MMDiT architecture design in *Stable Diffusion 3* (Esser et al., 2024). It supports bilingual generation and accommodates multiple tasks, including text-to-video and image-to-video synthesis, thereby providing a versatile foundation for building our framework.

## 3.2 MOTION INJECTION VIA IN-CONTEXT

Previous character animation approaches typically inject motion via auxiliary pose encoders, where motion is separately encoded and fused into the video backbone via cross-attention or feature concatenation. However, this additional modality branch shifts the backbone's inputs away from its large-scale pre-training distribution. The resulting large modality gap disrupts pre-trained generative priors, leading to degraded visual fidelity, weakened controllability, and poor generalization.

Inspired by *in-context learning* in large language models, where tasks are guided by directly concatenating prompts without altering the model structure, we adopt a structurally consistent strategy for motion-driven video generation. Specifically, we concatenate motion signals with the reference image at the spatial level. In this way, the backbone perceives motion as part of its *native input context*, rather than as an externally encoded feature. This design preserves the backbone's pre-training consistency, and allows LoRA adapters to seamlessly align motion and appearance in a plug-and-play manner without modifying the backbone architecture.

**In-Context Operation.** As shown in Fig. 2, our objective is to generate a video where the subject identity from a reference image $I_{\text{ref}} \in \mathbb{R}^{H \times W \times 3}$ faithfully follows the driving motion signals (poses or video sequences) $\mathbf{D} \in \mathbb{R}^{T \times H \times W \times 3}$. Central to our *in-context* design is the construction of a composite input sequence $\mathbf{C} \in \mathbb{R}^{T \times H \times 2W \times 3}$. For the first frame ($t = 1$), we spatially concatenate the reference image with the first frame of the driving video. For all subsequent frames ($t > 1$), we use a zero-filled image in place of the reference image. This process is formulated as:

$$\mathbf{C}[t] = \begin{cases} I_{\text{ref}} \oplus \mathbf{D}[t], & t = 1, \\ \mathbf{0} \oplus \mathbf{D}[t], & t > 1. \end{cases} \quad (2)$$

Here, $\oplus$ denotes spatial concatenation along the width axis, and $\mathbf{0}$ is a zero image matching the dimensions of $I_{\mathrm{ref}}$.

The composited video sequence $\mathbf{C}$ is fed into the video generation backbone to synthesize the target video. Initially, a 3D VAE compresses the video $\mathbf{C}$ into a latent sequence $\mathbf{Z}$. To accommodate the unique in-context input, which includes a dedicated region for motion signals, we construct a specialized mask condition. In detail, we create a motion mask $\mathbf{M}_m$ for highlighting the motion region, alongside a reference mask $\mathbf{M}_r$ following conventional image-to-video DiT models. Their spatially concatenation yields the final mask condition $\mathbf{M} = \mathbf{M}_r \oplus \mathbf{M}_m$. Finally, the video latent $\mathbf{Z}$, noise latent $\mathbf{Z}_{\mathrm{noise}}$, and the mask condition $\mathbf{M}$ are concatenated along the channel dimension. This combined representation is then passed to our video generation backbone for diffusion training, ultimately synthesizing the target video.

### 3.3 POSE-BASED DREAMACTOR-M2

We first present our **Pose-Based DreamActor-M2** framework, which adopts 2D pose skeletons as motion signals. The model is trained in a self-supervised manner. Given a reference video $\mathbf{V} \in \mathbb{R}^{T \times H \times W \times 3}$, we extract the character's pose sequence $\mathbf{P}$ to serve as the driving signal, and use the first frame of the video as the reference image $I_{\mathrm{ref}} = \mathbf{V}[0]$. Given image $I_{\mathrm{ref}}$ and pose sequence $\mathbf{P}$ as inputs, our model is tasked to reconstruct the original reference video $\mathbf{V}$. During inference, given a target character image and a driving video, our method can animate the character with consistent actions corresponding to the driving video.

**Pose Augmentation.** A key drawback of directly using 2D skeletons as motion signals is that they inevitably encode body-shape cues (e.g., limb length and body proportions). This can lead the animation model to overfit to these shape signals, compromising the preservation of the reference image's body shape and restricting *cross-identity* generation. To overcome this, we apply two *pose augmentation* strategies, which perturb skeleton representations to disrupt shape-dependent cues while retaining the essential motion dynamics:

**(1) Random Bone Length Scaling.** To further suppress body-shape cues, we introduce bone length scaling: bones are grouped into anatomical segments (e.g., arms, legs), each assigned a random scaling factor to adjust length proportionally. This ensures that while the overall pose dynamics are preserved, the skeletons exhibit diverse and randomized body proportions, preventing the model from overfitting to these shape signals.

**(2) Bounding-Box-Based Normalization.** We perform normalization on the skeleton sequence within each clip. We compute the bounding box that encloses all joints across the entire clip and normalize the joint coordinates to the size of this bounding box. This operation eliminates dependency on the subject's original position and scale, ensuring that skeletons from different identities are represented in a consistent, size-invariant manner.

**Text Guidance.** Pose augmentation, while effective in alleviating identity leakage, inevitably weakens motion semantics. For example, a "hands in prayer" motion might no longer resemble clasped hands after augmentation. To compensate for this, we propose a text guidance module that provides richer and high-level motion semantic context. As depicted in Fig. 2, a pre-trained MLLM extracts *motion semantics* from the driving video $\mathbf{V}$ and *appearance semantics* from the reference image $I_{\mathrm{ref}}$. Subsequently, these semantics are fused by an LLM into a unified, target-oriented description. The resulting textual representation $T_{\mathrm{fusion}}$ is subsequently fed into the video generation model through its text-conditioning module. This high-level semantic guidance complements low-level pose signals, enhancing both the controllability and expressiveness of character animation.

**LoRA Fine-tuning.** Since the video generation backbone has been pretrained on large-scale datasets, it inherently encodes strong generative priors, such as body proportion preservation and motion consistency. To retain these capacities, we adopt a parameter-efficient adaptation strategy based on LoRA (Hu et al., 2022) fine-tuning. Specifically, all backbone parameters are frozen, and lightweight LoRA modules are inserted only into the feed-forward layers (excluding text cross-attention). Because both in-context images and textural captions are native input modalities, our framework achieves seamless integration without any architectural modification, enabling plug-and-play adaptation to motion control tasks.

## 3.4 End-to-End Training Paradigm

While Pose-based DreamActor-M2 demonstrates high-quality motion transfer from human videos, its reliance on pose estimation limits robustness in complex scenarios or non-human cases. To overcome this limitation, we extend the framework to an *end-to-end variant* that directly leverages raw RGB frames as motion signals, referred to as **End-to-End DreamActor-M2** (Fig. 2). The overall training paradigm is divided into two stages: (i) data synthesis and quality filtering (details are provided in Appendix D), and (ii) model optimization.

**Data Synthesis and Quality Filtering.** A fundamental obstacle to end-to-end training is the absence of large-scale paired data that simultaneously ensures motion consistency and cross-identity diversity. To address this challenge, we introduce a **self-bootstrapped data synthesis pipeline** that leverages the strong motion transfer capability of our pre-trained *Pose-based DreamActor-M2* to automatically construct pseudo-paired supervision.

Formally, given a source driving video $\mathbf{V}_{\text{src}}$, we first extract its pose sequence $\mathbf{P}_{\text{src}}$ to serve as the motion control signals. This pose sequence, in conjunction with a different-identity image $I_o$, is then fed into the pose-based DreamActor-M2 (denoted as $\mathcal{M}_{\text{pose}}$) to synthesize a motion-consistent yet identity-diverse video $\mathbf{V}_o$:

$$\mathbf{V}_o = \mathcal{M}_{\text{pose}} \left( \mathbf{P}_{\text{src}}, I_o \right), \tag{3}$$

The synthesized $\mathbf{V}_o$ retains the motion dynamics of $\mathbf{V}_{\text{src}}$ while featuring a new identity, forming an initial pseudo-pair $(\mathbf{V}_{\text{src}}, \mathbf{V}_o)$. Next, an MLLM assesses the quality and semantic alignment of the synthesized pairs, filtering out samples with poor fidelity. This quality-control step ensures that only reliable and semantically coherent pairs are retained for downstream training.

**Model Optimization.** With this curated pseudo-paired data, we repurpose $\mathbf{V}_o$ as the driving video, and use the first frame of $\mathbf{V}_{\text{src}}$ as the reference image, denoted as $I_{\text{ref}} = \mathbf{V}_{\text{src}} [0]$. This process yields a large-scale, high-quality pseudo-paired dataset:

$$\mathcal{D} = \{ (\mathbf{V}_o, I_{\text{ref}}, \mathbf{V}_{\text{src}}) \}. \tag{4}$$

Each triplet serves as a training instance where $\mathbf{V}_o$ provides the motion context, $I_{\text{ref}}$ specifies the target identity, and $\mathbf{V}_{\text{src}}$ acts as the ground-truth supervision. We then use this dataset $\mathcal{D}$ to train our end-to-end DreamActor-M2, optimizing it to reconstruct $\mathbf{V}_{\text{src}}$ from the inputs $(\mathbf{V}_o, I_{\text{ref}})$.

This training paradigm enables the network to learn motion transfer directly from raw RGB inputs, thereby eliminating the reliance on explicit pose annotations. The scalable and annotation-free supervision facilitates robust end-to-end training. To further accelerate convergence and stabilize training, we warm-start the end-to-end framework using the pre-trained weights of the Pose-Based DreamActor-M2. This initialization allows the model to inherit strong motion transfer priors, significantly improving optimization efficiency and enhancing final performance.

Together, the End-to-End DreamActor-M2 and its pose-based counterpart constitute a unified framework for character image animation, supporting both end-to-end training and inference. To the best of our knowledge, this is the first fully end-to-end solution for this task, representing a substantial step toward practical and scalable character animation systems.

## 4 Experiments

### 4.1 Implementations

**Implementation Details.** Following (Hu et al., 2023; Zhang et al., 2024), we use DWPose (Yang et al., 2023) to extract pose sequences from videos and render them as OpenPose-style skeleton images (Cao et al., 2017). We collect 100,000 human videos from the Internet for training, randomly sampling 49–121 frames from each video. Given the strong multimodal understanding and language integration capabilities of Gemini 2.5 (Comanici et al., 2025), we adopt it as both the MLLM and LLM in our framework. All experiments are conducted on 16 H20 GPUs. The training involves 50,000 steps with a batch size of 2. Our generative backbone is the pretrained image-to-video DiT model seedance 1.0 (Gao et al., 2025), with the LoRA rank set to 256. We use AdamW optimizer with a learning rate of 5e-5 and a weight decay of 0.01. During inference, each generated video segment contains 121 frames (about 5 seconds). The details regarding data synthesis and filtering for training end-to-end DreamActor-M2 are provided in Appendix D.

Table 1: Quantitative comparisons with SOTAs on `AWBench`.

| Method | Video-Bench↑ | | | | | |
|---|---|---|---|---|---|---|
| | Imaging Quality | Aesthetic Quality | Temporal Consistency | Motion Smoothness | Background Consistency | Subject Consistency |
| MimicMotion (Zhang et al., 2024) | 3.05 | 2.67 | 3.26 | 2.83 | 2.02 | 2.16 |
| DisPose (Li et al.) | 3.38 | 2.91 | 3.50 | 3.03 | 2.67 | 2.72 |
| Animate-X (Tan et al., 2024) | 4.03 | 3.48 | 3.84 | 3.69 | 3.25 | 3.01 |
| Unianimate-DiT (Wang et al., 2025) | 4.16 | 3.67 | 3.92 | 3.96 | 4.02 | 3.75 |
| MTVCrafter (Ding et al., 2025) | 4.21 | 3.50 | 4.02 | 4.11 | 3.91 | 3.83 |
| Animate-X++ (Tan et al., 2025) | 4.25 | 3.92 | 4.15 | 4.02 | 3.97 | 3.91 |
| DreamActor-M1 (Luo et al., 2025) | 4.57 | 4.28 | 4.31 | 4.29 | 4.38 | 4.21 |
| Ours (Pose-based DreamActor-M2) | **4.68** | **4.76** | **4.61** | **4.53** | **4.74** | **4.38** |
| Ours (End-to-End DreamActor-M2) | **4.72** | **4.78** | **4.69** | **4.56** | **4.68** | **4.35** |

**Evaluation Benchmark.** To comprehensively evaluate the efficacy and generalizability of our proposed DreamActor-M2, we curated a dedicated benchmark `AWBench` encompassing a wide range of motion types and reference identities. The benchmark consists of 100 driving videos and 200 reference images, where the driving corpus covers human as well as non-human motion categories (see Appendix E for detailed dataset construction).

**Evaluation metrics.** Most evaluation metrics, such as FID-FVD (Balaji et al., 2019), FVD (Unterthiner et al., 2018), and CD-FVD (Ge et al., 2024), rely on comparisons with ground-truth videos, which are unavailable in cross-identity animation scenarios. As a result, these metrics fail to accurately reflect model performance. Moreover, prior studies have revealed that these metrics are often inconsistent with human judgment (Huang et al., 2024). To address this, we adopt the **human-aligned evaluation metrics** in Video-Bench (Han et al., 2025), which assesses generation quality across six perceptual dimensions: **imaging quality**, **aesthetic quality**, **temporal consistency**, **motion smoothness**, **background consistency**, and **subject consistency**. These dimensions offer a more reliable and comprehensive evaluation of character animation models in real-world settings. For each dimension, Video-Bench will automatically assign a score, with the scores and their corresponding grades as follows: 1-very poor, 2-poor, 3-moderate, 4-good, 5-excellent.

## 4.2 QUANTITATIVE COMPARISON

We evaluate the model via `AWBench`. Given other SOTA methods only support human driving videos, we select 60 human driving video-human reference image pairs and 40 human driving video-cartoon reference image pairs from `AWBench` as the quantitative test set. Lacking ground-truth for generated results, we adopt Video-Bench (Han et al., 2025) for evaluation, with results in Tab. 1. Our method achieves the best performance across all dimensions, verifying its effectiveness in generation quality and generalization. In Aesthetic Quality, End-to-End DreamActor-M2 scores the highest (4.78), significantly outperforming other methods and highlighting its strength in subjective visual appeal. For Subject Consistency, both Pose-based (4.38) and End-to-End (4.35) DreamActor-M2 rank top, with scores notably higher than MimicMotion (2.16), ensuring stable subject appearance during animation generation.

## 4.3 QUALITATIVE RESULTS

**Comparison with State-of-the-arts.** Fig. 3 presents qualitative comparisons on our benchmark, focusing on cross-identity animation. The first row shows our DreamActor-M2 excels in image quality, identity preservation, and motion alignment. The second row highlights its superior body shape preservation and facial expression alignment with driving inputs, outperforming others in capturing consistent expression details. The third row demonstrates its ability to accurately generate the "heart gesture" (a feat other methods fail to achieve), validating superior motion semantic capture. The fourth row underscores its advantage in multi-subject driving scenarios where competitors falter. Overall, DreamActor-M2 outperforms SOTAs in identity preservation and motion consistency.

**Adaptability across Human and Portrait Images.** Fig. 4 (a) shows DreamActor-M2's strong adaptability in human-portrait motion transfer, excelling in human-to-portrait, portrait-to-portrait, and portrait-to-human animations. For portrait-to-human cases, the far-right example maintains

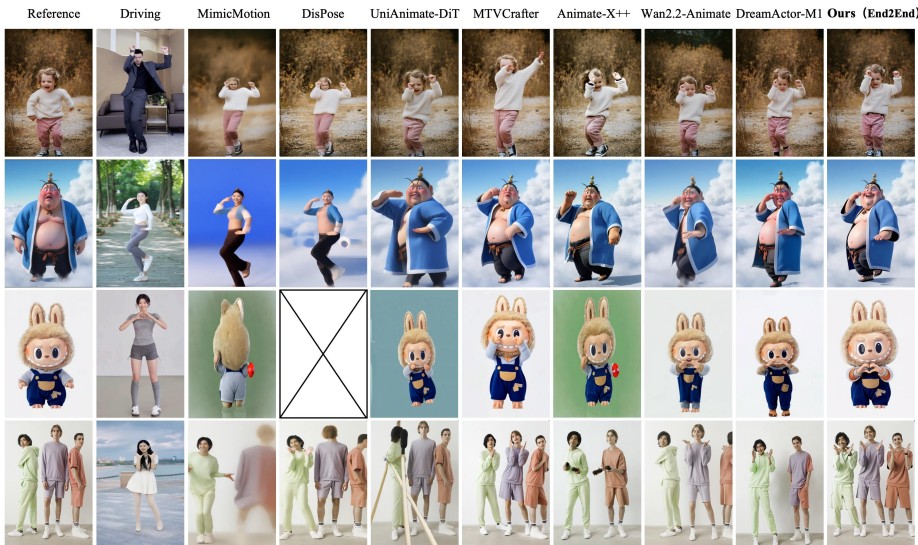

Figure 3: Qualitative comparisons between our method and state-of-the-art approaches on *AW*Bench.

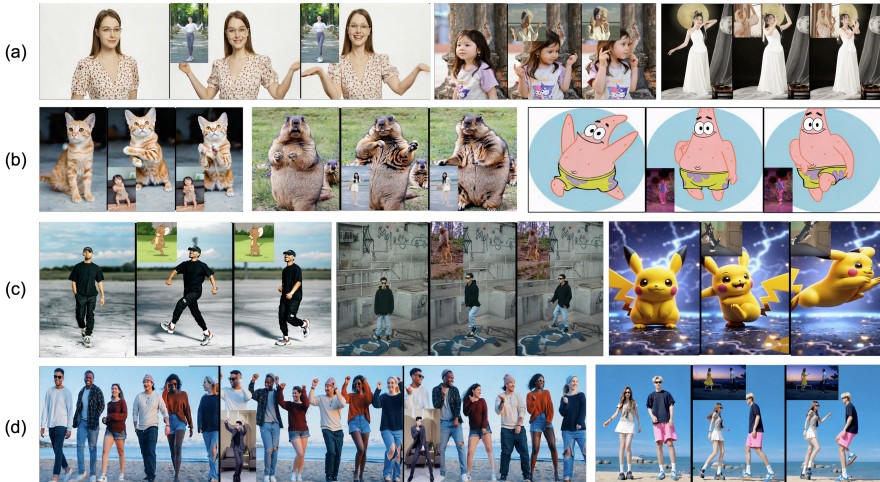

Figure 4: Qualitative visualization for various scenarios, such as various shot types (a), reference character types (b), driving character types (c), and multi-person settings (d).

coherence between the uncontrolled lower body and controlled upper body without visual distortions or artifacts.

**Adaptability to Non-Human Reference Image.** Unlike most methods limited to human animation, our approach adapts to diverse references (humans, animated characters, animals, non-humanoids). Fig. 3 shows effective animation of humans/humanoids with preserved body shape and appearance. Fig. 4 (b) further demonstrates precise animation of non-humans (cat, marmot) and non-humanoid anime characters, ensuring high-fidelity appearance and pose consistency with driving videos.

**Adaptability to Non-Human Driving Videos.** Unlike methods restricted to human driving videos (relying on explicit pose control), our end-to-end DreamActor-M2 directly conditions on video input, supporting non-human motion sources. Fig. 1 validates bird-to-bird animation, while Fig. 4 (c) shows effective handling of non-human animated/animal driving videos, enabling non-human-to-human and non-human-to-non-human scenarios, confirming broad generalization beyond human-centric settings.

Table 2: Ablation study on proposed DreamActor-M2 framework.

| Model | Method | Video-Bench↑ | | | | | |
|---|---|---|---|---|---|---|---|
| | | Imaging Quality | Aesthetic Quality | Temporal Consistency | Motion Smoothness | Background Consistency | Subject Consistency |
| A | Temporal-level in-context | 4.52 | 4.50 | 4.46 | 4.38 | 4.71 | 4.31 |
| B | w/o pose normalization | 4.58 | 4.64 | 4.57 | 4.46 | 4.66 | 4.27 |
| C | w/o pose bone length rescale | 4.57 | 4.66 | 4.55 | 4.63 | 4.58 | 4.11 |
| D | w/o any pose augmentation | 4.55 | 4.62 | 4.53 | 4.41 | 4.64 | 3.98 |
| E | w/o target-oriented description | 4.60 | 4.63 | 4.51 | 4.28 | 4.50 | 4.02 |
| F | Text cat w/o LLM | 4.65 | 4.70 | 4.56 | 4.41 | 4.66 | 4.27 |
| G | Full Model (Pose-based DreamActor-M2) | **4.68** | **4.76** | **4.61** | **4.53** | **4.74** | **4.38** |

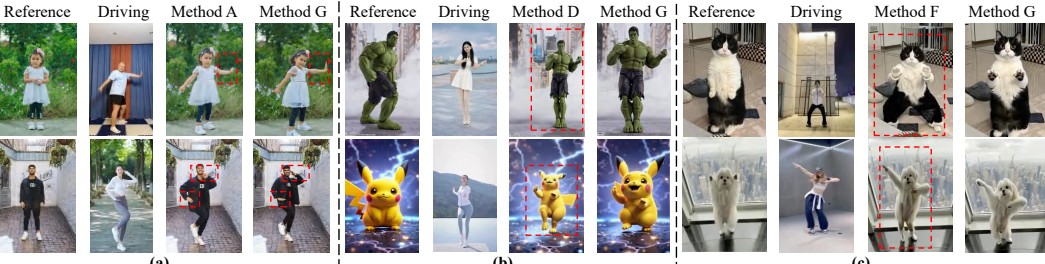

Figure 5: Qualitative visualization for ablation study.

**Adaptability to Multi-Person Driving.** Fig. 3 confirms our superiority in single-to-multi-person animation. Fig. 4 (d) demonstrates precise motion control for six characters (left) and successful multi-to-multi animation (right, transferring distinct motions from two driving subjects to reference positions).

## 4.4 ABLATION STUDY

In this section, we conduct ablation studies to demonstrate the effectiveness of each component of DreamActor-M2. The results are summarized in Tab. 2 and Fig. 5. Firstly, we compare the original spatial injection approach (Model G) with the temporal in-context method (Model A). As shown in Tab. 2, spatial injection yielded better overall generation quality. Moreover, as illustrated in Fig. 5 (a), it preserves finer hand details, demonstrating its advantage in spatial fidelity. Method G also exhibits better performance in hand motion alignment than Method A. Next, by comparing Models B, C, D, and G in Tab. 2, we observe that pose augmentation leads to further quality improvements. Fig. 5 (b) can illustrate that models incorporating pose augmentation better maintain the body shape of the original reference image. Finally, by comparing Models E, F, and G in Tab. 2, we find that injecting target motion text information through the LLM yields superior results. Fig. 5 (c) further indicates that models guided by target-oriented descriptions exhibit a better balance between reference image character preservation and motion consistency.

## 5 CONCLUSION

We introduce DreamActor-M2, a universal framework designed for character image animation. A key innovation of our method is an align-free in-context LoRA strategy that unifies motion signals and reference images into a single input representation. This design not only preserves the powerful generative capabilities of the pretrained backbone, but also enables end-to-end motion transfer directly from raw driving videos, bypassing the need for explicit pose estimation. The versatility of the framework enables its application to a wide range of characters, encompassing both human and non-human subjects. Our extensive experiments confirm that DreamActor-M2 achieves high effectiveness, strong generalization, and exceptional versatility. These results collectively establish a unified and powerful paradigm for character animation across a variety of scenarios.

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

## A    ETHICS STATEMENT

This work adheres to the ICLR Code of Ethics. In this study, no human subjects or animal experimentation was involved. All datasets used were sourced in compliance with relevant usage guidelines, ensuring no violation of privacy. We have taken care to avoid any biases or discriminatory outcomes in our research process. No personally identifiable information was used, and no experiments were conducted that could raise privacy or security concerns. We are committed to maintaining transparency and integrity throughout the research process.

## B    REPRODUCIBILITY STATEMENT

We have made every effort to ensure that the results presented in this paper are reproducible. Code will be made publicly available to facilitate replication and verification after inspection. The experimental setup, including training steps, model configurations, and hardware details, is described in detail in the paper. We believe these measures will enable other researchers to reproduce our work and further advance the field.

## C    LLM USAGE

Large Language Models (LLMs) were used to aid in the writing and polishing of the manuscript. Specifically, we used an LLM to assist in refining the language, improving readability, and ensuring clarity in various sections of the paper. The model helped with tasks such as sentence rephrasing, grammar checking. It is important to note that the LLM was not involved in the ideation, research methodology, or experimental design. All research concepts, ideas, and analyses were developed and conducted by the authors. The contributions of the LLM were solely focused on improving the linguistic quality of the paper, with no involvement in the scientific content or data analysis. The authors take full responsibility for the content of the manuscript, including any text generated or polished by the LLM. We have ensured that the LLM-generated text adheres to ethical guidelines and does not contribute to plagiarism or scientific misconduct.

## D    DATA SYNTHESIS AND QUALITY FILTERING DETAILS

In this section, we present the implementation details of the data synthesis and quality filtering stage in training End-to-End DreamActor-M2 framework.

Due to the scarcity of large-scale, high-quality paired video datasets that simultaneously satisfy both cross-identity and motion-consistency requirements, we employed a pretrained pose-based DreamActor-M2 model to synthesize data for our end-to-end training pipeline. We first curated a dataset comprising 10,000 human driving videos with diverse motion patterns and 200 high-resolution reference images representing a variety of subjects, including humans, anime characters, and animals. During the synthesis stage, we randomly extracted a 5-second (121-frame) segment from each driving video as the motion signal and randomly selected one reference image. This process generated a 5-second identity-crossing video, and we stored the driving clip, synthesized video, and reference image together as a single training unit, ultimately producing 600,000 training units for end-to-end training.

The raw synthesized videos required additional filtering for effective training. To this end, we employed the Video-Bench (Han et al., 2025) evaluation model to perform automatic quality assessment. Each video was evaluated on a 5-point scale (1: very poor, 5: excellent) across six dimensions: imaging quality, aesthetic quality, temporal consistency, motion smoothness, background consistency, and subject consistency. We then calculated the average score for each video across these dimensions and retained only those training units with an average score greater than 4 for inclusion in the final end-to-end training dataset.

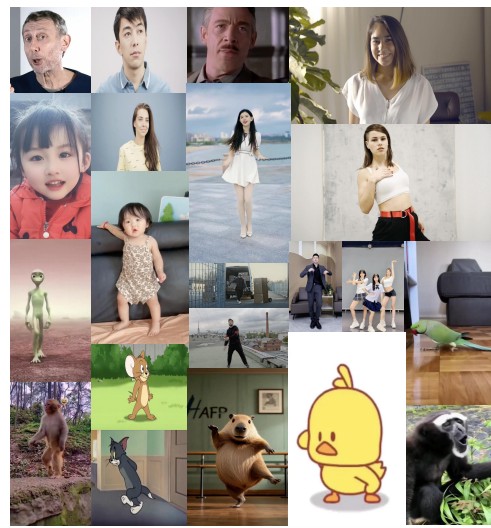 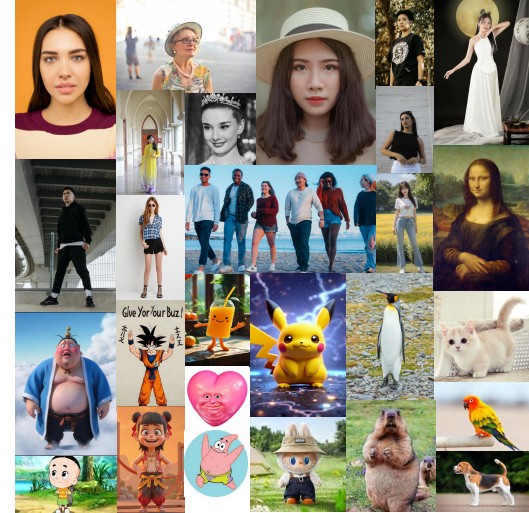

(a) Driving video examples from our *AW*Bench      (b) Reference image examples from our *AW*Bench

Figure 6: Visual examples of driving video corpus and reference image corpus.

# E CONSTRUCTED EVALUATION BENCHMARK

To thoroughly evaluate the effectiveness and generalizability of our proposed DreamActor-M2, we constructed a comprehensive evaluation benchmark `AWBench` encompassing diverse motion types and reference identities. The benchmark consists of both driving videos and reference images. The detailed statistics of `AWBench` are shown in Tab. 3. The driving video corpus covers a wide range of human and non-human motions. Human motions are sampled across different body regions (face, upper body, full body), age groups (child, young adult, elderly), and activity categories (e.g., dancing, daily activities), and include both camera-tracked and static-camera sequences. Non-human motions include videos of animals (such as cats, chickens, parrots, monkeys, and orangutans) and animated characters (such as Tom the cat, Jerry the mouse, groundhogs, and cartoon aliens). In total, the evaluation benchmark contains 100 driving videos and 200 reference characters. Visual examples of the driving video corpus and reference images are shown in Fig. 6(a) and Fig. 6(b), respectively.

# F EVALUATION FOR LONG GENERATED VIDEOS

In this section, we present several long video visualization results generated based on DreamActor-M2. The lengths of the three examples (A), (B), and (C) are 16s, 20s, and 24s respectively, while the length of the training samples we used is 5s. In Fig. 7, we can observe that the videos generated by our method can effectively maintain human identity, ensure excellent video frame quality, and achieve good motion coherence of the characters.

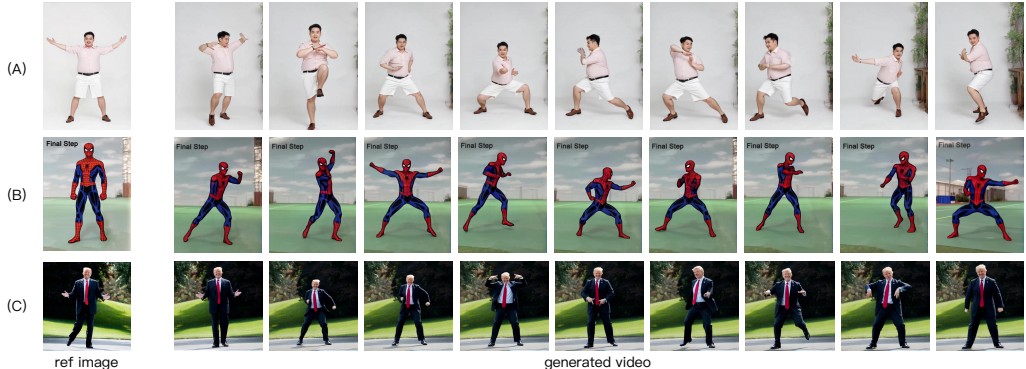

ref image      generated video

Figure 7: Qualitative visualizations for generated long videos based on DreamActor-M2.

Table 3: Detailed Statistics of the `AWBench`

| Category | Sub-category | Specific Items | Count | Ratio/Note |
|---|---|---|---|---|
| **Overall Scale** | Driving Videos | – | **100** | 100% |
| | Reference Images (Identities) | – | **200** | 100% |
| **Human Videos** | **Total** | – | 75 | **75%** |
| | By Body Region | Facial Motions | 25 | 33.3% of Human |
| | | Upper-body Motions | 25 | 33.3% of Human |
| | | Full-body Motions | 25 | 33.3% of Human |
| | By Age Group | Child | 20 | 26.7% of Human |
| | | Young Adult | 35 | 46.7% of Human |
| | | Elderly | 20 | 26.7% of Human |
| | By Activity Type | Dancing | 25 | 33.3% of Human |
| | | Daily Activities | 35 | 46.7% of Human |
| | | Other Professional Actions | 15 | 20.0% of Human |
| | By Camera Type | Camera-tracked Sequences | 40 | 53.3% of Human |
| | | Static-camera Sequences | 35 | 46.7% of Human |
| **Non-human Videos** | **Total** | – | 25 | **25%** |
| | Animals | Cats, Chickens, Parrots, Monkeys, Orangutans, etc. | 15 | 60% of Non-human |
| | Animated Characters | Tom the cat, Jerry the mouse, Groundhogs, Cartoon aliens, etc. | 10 | 40% of Non-human |
| **Reference Images** | By Category | Human Identities | ~150 | ~75% of Total Ref. |
| | | Non-human Identities | ~50 | ~25% of Total Ref. |

## G Prompt engineering for MLLM and LLM

Leveraging the robust multimodal comprehension capabilities of Gemini 2.5 Pro, we employ it as both our Multimodal Large Language Model (MLLM) and Large Language Model (LLM) in this study. The core innovation lies in the design of specialized prompts to guide the model in extracting task-specific, disentangled semantic information from raw inputs.

To enable Gemini 2.5 Pro to extract *appearance-agnostic motion semantics* from raw driving videos, we designed a specific prompt, termed the *motion prompt*. The motion prompt is defined as follows:

Listing 1: The Prompt for Extracting Motion Information for driving video

```
input_motion_prompt = f"""Analyze the character's motion in the video and
    generate a detailed motion description strictly per the following
    requirements (no line breaks in the final output).
1. Mandatory Basic Constraints
Scope: Only describe motion, pose, expression and movement changes;
    exclude appearance (clothing, hair color, etc.) and gender.
Completeness: Cover all motion details of face, hands and body; no
    omission of key pose changes.
Accuracy: Precisely describe hand orientation and relative positions;
    avoid ambiguity (e.g., specify direction/position instead of "hand
    raised").
2. Core Description Requirements
Time Range: Focus solely on the first 5 seconds of the video.
Hand Details: Explicitly specify for both hands: back orientation (self/
    viewer/left/right), position relative to body parts, and object/
    character interaction (if applicable).
Expressions: Track dynamic facial expression changes (e.g., "smile to
    laughter").
Auxiliary Behaviors: Include behaviors like speaking if present.
3. Fixed Output Structure
Opening: Start with "In the video, the character's initial pose is".
Temporal Transition: Use in order: "initial pose is" (start), "
    Subsequently," (1st change), "Then," (2nd change), "After that," (3rd
     if applicable), "Finally," (final state).
Format: Single continuous paragraph; no line breaks, bullet points or
    subheadings.
4. Example Reference
In the video, the character's initial pose is smiling, raising his right
    hand parallel with his face, left hand gripping another person's arm;
     right palm faces himself, back of right hand faces others as if
    showing a ring. Subsequently, he lowers his right hand. Then, he
    continues speaking with left hand unchanged. After that, his
    expression softens. Finally, he stops speaking and maintains a gentle
     smile."""
```

To guide Gemini 2.5 Pro in extracting *pose-agnostic appearance information* from reference images, we developed a structured appearance prompt. This prompt strictly defines the scope of appearance description, excludes irrelevant motion/behavioral content, and standardizes the output format, ensuring the model captures comprehensive and accurate visual attributes of the character. The full prompt is defined as follows:

Listing 2: The Prompt for Extracting Appearance Information for referring image

```
input_appearance_prompt = f"""Analyze the character's visual attributes
    in the image and generate a detailed appearance description strictly
    per the requirements below (no line breaks in the final output).
1. Mandatory Basic Constraints
Scope: Describe only appearance and gender; exclude actions, posture,
    movement (e.g., "raising hands", "sitting posture" are prohibited).
```

```
Completeness: Cover all visible appearance details; no omission of key
    attributes (clothing, accessories, facial features).
Accuracy: Align description with image content; avoid subjective
    speculation (e.g., describe as "long hair, gender undistinguishable"
    if unclear).
2. Core Description Scope
Gender: Specify (male/female/gender undistinguishable) based on visual
    cues.
Facial Features: Detail hair (color/style), eyes, eyebrows, lip color,
    and facial accessories (glasses/necklaces).
Clothing & Accessories: Describe clothing style/color/details and
    accessories (watch/bag) clearly.
Held Objects: Explicitly include all objects held by the character.
Body Features: Describe visible attributes (skin tone, body shape)
    without involving posture/movement.
3. Output Format Specification
Unity: Single continuous paragraph; no line breaks, bullet points, or
    subheadings.
Logic: Organize in coherent sequence (e.g., gender to facial features to
    clothing to held objects).
Conciseness: Avoid redundant repetitions of the same attribute.
4. Example Reference
The character in the image is female with chest-length brown curly hair,
    thick black eyebrows, and light pink lip gloss. She wears a white
    short-sleeved blouse with small blue polka dots, a gray midi skirt,
    and silver stud earrings. Her left hand holds a printed beige canvas
    bag, right hand grips a white book with a blue cover; she has fair
    skin and a thin silver bracelet on her right wrist."""
```

Finally, leveraging the robust linguistic capabilities of Gemini 2.5 Pro, we fuse the generated textual descriptions of motion and appearance—specifically, transferring the motion described in the motion text to the character (as depicted in the appearance text)—while ensuring the fused text aligns with the character's identity as specified in the appearance description. Our structured fusion prompt template is defined as follows:

Listing 3: The Prompt for Fusing Motion Information with Appearance Information

```
input_fusion_prompt = f"""Strictly follow the requirements below to
    transfer the character's motion from the video to the image's
    character appearance, and generate a complete, natural, fluent action
     description (no line breaks in output).
1. Mandatory Basic Constraints
Appearance: Fully retain all details from appearance_caption; no omission
     or modification.
Motion: Fully retain all motion, posture, expression and behavior from
    motion_caption; no omission or modification.
Rationality: Comply with biological common sense; avoid unreasonable body
     structures (e.g., bird wings have no palms/fingers).
Fusion Method: Only perform reasonable body part mapping to match motion
    with appearance; no tampering with original content.
2. Character Quantity Rules
Base on the number of characters in the image.
Video multiple characters + Image 1 character: Assign motion of the video
    's most prominent character.
Image multiple characters + Video 1 character: Assign video motion to all
     image characters.
Inconsistent quantity: Prioritize retaining image characters' appearance
    and quantity while fully preserving video motion.
3. Non-Human Character Motion Mapping Rules (Apply only if image
    character is non-human)
Human to Bird: Hands/Arms to Wings (palms/fingers substituted with wing
    edge/direction/opening-closing); Legs/Foot to Claws.
```

```
Human to Feline/Quadruped: Hands to Forelegs; Legs to Hind legs; Palm
    alignment to Forepaw alignment.
Other Cases: Upper limbs to Forelimbs/tactile appendages; Lower limbs to
    Hind limbs/supporting limbs; Fingers/Palms to Organ-specific motions
    (wing flapping, claw opening).
General Rule: Ignore unmatched body part details; replace with reasonable
    descriptions; no redundant anatomical structures.
Environment Interaction: Fully retain video interactions and map
    logically to image character's structure.
4. Output Requirements
Integrate all content from appearance_caption and motion_caption; no
    omission/modification.
Ensure reasonable, accurate motion transfer and natural integration with
    appearance.
Fluent and coherent language; no awkward splicing.
For bird characters: Exclude irrelevant terms (human, hand, palm, fist,
    etc.).
5. Input
Image Appearance (appearance_caption): {appearance_caption}
Video Motion (motion_caption): {motion_caption}
6. Output (Start with "In the video, a/an...")
Generate a complete natural language text integrating the above
    appearance and motion.
7. Example
Input:
appearance_caption: A young woman with dark hair in a high bun (adorned
    with purple flowers, gold accents), slender eyebrows, large bright
    eyes with winged eyeliner, light pink lip gloss, fair skin, blush.
    She wears a black kimono (multi-layered pink-red neckline, butterfly/
    flower patterns on right shoulder/sleeves) and a gold-patterned white
     waist belt.
motion_caption: Initially smiling, raises right hand parallel to face (
    palm facing self, back facing others, as if showing a ring), left
    hand grabs another's arm. Lowers right hand, speaks, expression
    shifts from smile to laughter.
Output:In the video, a young woman has dark hair in a high bun adorned
    with purple flowers and gold accents, slender eyebrows, large bright
    eyes with winged eyeliner, light pink lip gloss, fair skin and blush.
     She wears a black kimono with a multi-layered pink-red neckline,
    butterfly and flower patterns on her right shoulder and sleeves, and
    a gold-patterned white waist belt. Initially, she smiles, raises her
    right hand parallel to her face (palm facing self, back facing others
    , as if showing a ring), and her left hand grabs another person's arm
    . Subsequently, she lowers her right hand, speaks, and her expression
     gradually changes from a smile to laughter."""
```

## H    MLLM FOR APPEARANCE AND MOTION UNDERSTANDING

In this section, we present partial qualitative visualization results of appearance descriptions and motion descriptions generated for reference images and driving videos respectively based on Gemini 2.5 Pro. First, we randomly selected 100 reference images and 100 driving videos from our training samples. Then, we used the designed prompts for guidance, and Gemini 2.5 Pro generated the corresponding appearance descriptions and motion descriptions. Finally, human evaluators scored the generated appearance and motion descriptions again using a 5-point scale (1 = very poor, 2 = poor, 3 = moderate, 4 = good, 5 = excellent). According to statistics, among the selected samples, the proportion of generated appearance descriptions receiving a human evaluation score of over 4 reaches as high as 98%, and the proportion of generated motion descriptions with a score of over 4 is as high as 97%. We present 5 reference images along with their appearance descriptions generated by Gemini 2.5 Pro, 3 driving videos along with their motion descriptions generated by Gemini 2.5 Pro, and their corresponding human evaluation scores in Fig. 8 and Fig. 9. Based on these

quantitative and qualitative results, we can conclude that the appearance descriptions and motion descriptions generated by Gemini 2.5 Pro are quite reliable and stable.

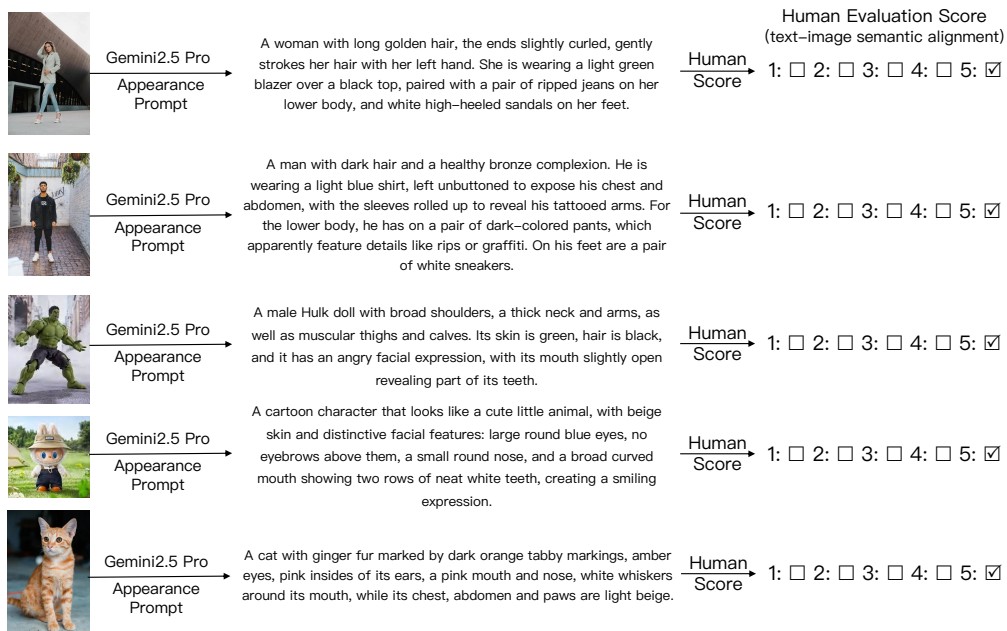

Figure 8: Qualitative visualizations for generating appearance caption based on Gemini2.5 Pro.

# I MLLM FOR EVALUATING AND FILTERING SYNTHESIZED VIDEOS

In this section, we qualitatively evaluated the reliability of the MLLM Gemini2.5 Pro in assessing and filtering synthesized video pairs. The evaluation focused on subject consistency between the generated video and reference image, as well as motion consistency with the driving video. We adopted a 5-point scale (1 = very poor, 2 = poor, 3 = moderate, 4 = good, 5 = excellent) for Gemini2.5 Pro to automatically score via the Video-Bench pipeline. To validate its reliability for synthesized data evaluation, we described the appearance of a random frame from both the reference image and generated video to judge subject consistency, and semantically described motions of the driving and generated videos to determine motion alignment. Fig. 11 demonstrates that the MLLM effectively evaluates synthesized paired data, with reliable filtering results.

# J EDGE VIDEO CASES FOR MLLM

In this section, we present visualizations of several edge cases—illustrated in Figure 1—where accurately extracting motion semantics poses notable challenges for the MLLM.

(A) Occluded/Small Subjects: When the moving subject occupies a small spatial region or is partially obscured, the MLLM struggles to capture its detailed motion semantics due to insufficient visual cues.

(B) Camera Motion Artifacts: Significant camera shake or unstable motion introduces ambiguous global motion signals, degrading the MLLM's ability to distinguish subject-specific motion from camera-induced artifacts.

(C) High-Speed Motion: In scenes with rapid movement, motion blur and frame-to-frame discontinuity reduce the clarity of motion patterns, limiting the MLLM's comprehension of fine-grained dynamics.

(D) Heavy Occlusion: Substantial occlusion of the moving subject results in incomplete visual evidence, preventing the MLLM from inferring accurate and consistent motion semantics.

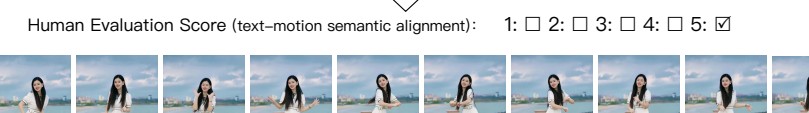

Gemini2.5 Pro → Motion Prompt

A person starts with their arms naturally hanging down, head held straight, and a calm facial expression. Subsequently, they lift both arms to the front of their chest, clench their hands into fists with palms facing themselves, while tilting their head slightly to the left and wearing a faint smile. Immediately after, they stretch their right arm upward, with the index and middle fingers together pointing upward, and their left arm downward straight, with the index and middle fingers together pointing downward, maintaining the smile and staring straight ahead. Then, they raise both arms upward, with the backs of their hands facing themselves, the index and middle fingers spread apart, and the remaining fingers closed, forming a "V" shape with both hands; at the same time, they lean their head forward and look ahead. Finally, they spread both arms sideways, with the backs of their hands still facing themselves, the index and middle fingers spread apart, and the remaining fingers closed, keeping the hands in a "V" shape, while the head position and facial expression remain the same as in the previous movement.

Human Score ⬇

Human Evaluation Score (text–motion semantic alignment): 1: ☐ 2: ☐ 3: ☐ 4: ☐ 5: ☑

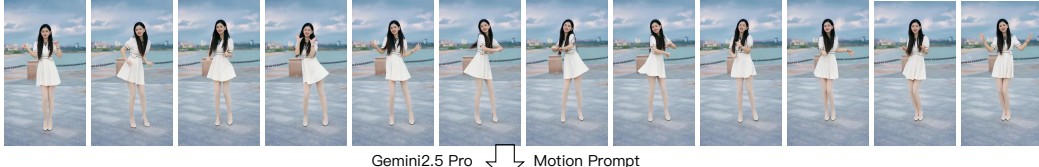

Gemini2.5 Pro → Motion Prompt

A person stands frontally initially, with the body tilting slightly to the left. Both hands are raised to chest height, each formed into a half-heart shape—backs of the hands facing the audience, fingertips pointing obliquely upward—while wearing a smile. Subsequently, the person lowers both hands, spreads the palms open and stretches them flat downward (backs still facing the audience), and slides the legs apart to both sides simultaneously. Next, the right leg is retracted, the upper body tilts slightly leftward again, and both hands form half-heart shapes once more (backs facing the audience, fingertips pointing obliquely upward) with the smile maintained. Then, the person lowers the hands again, spreads the palms open and stretches them flat downward (backs facing the audience), while sliding the legs apart to both sides. Finally, the right leg is retracted, the upper body tilts slightly to the left, and both hands form half-heart shapes again—backs facing the audience, fingertips pointing obliquely upward—keeping the smile on the face.

Human Score ⬇

Human Evaluation Score (text–motion semantic alignment): 1: ☐ 2: ☐ 3: ☐ 4: ☐ 5: ☑

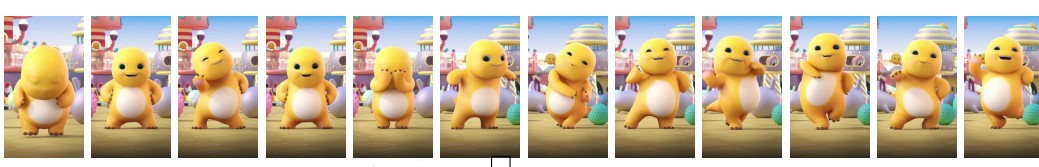

Gemini2.5 Pro → Motion Prompt

The character's initial posture is with hands on the head, head bowed downward, eyes closed, performing a peekaboo gesture. Subsequently, the character releases their hands, opens their eyes, and looks at the audience with a smile. Next, the character raises their right hand to be parallel to the body, palms open, then lowers the right hand while maintaining eye contact with the audience. Then, the character places their hands on the head again, bows their head downward, closes their eyes, and repeats the peekaboo gesture. Finally, the character lowers their hands once more, looks at the audience with a smile, and tilts their body to the right. After that, the character stands upright, lifts their right foot, keeps both hands at their sides, and tilts their head to the left, then looks at the audience with a smile. The character then raises their left hand to chest level, palms open, while placing the right hand on the right side of the body. Subsequently, the character brings their legs together, places the left hand on the left side of the body, raises the right hand to be parallel to the head, and continues smiling at the audience. Eventually, the character turns around to face away from the audience, with both hands hanging naturally at their sides.

Human Score ⬇

Human Evaluation Score (text–motion semantic alignment): 1: ☐ 2: ☐ 3: ☐ 4: ☐ 5: ☑

Figure 9: Qualitative visualizations for generating motion caption based on Gemini2.5 Pro.

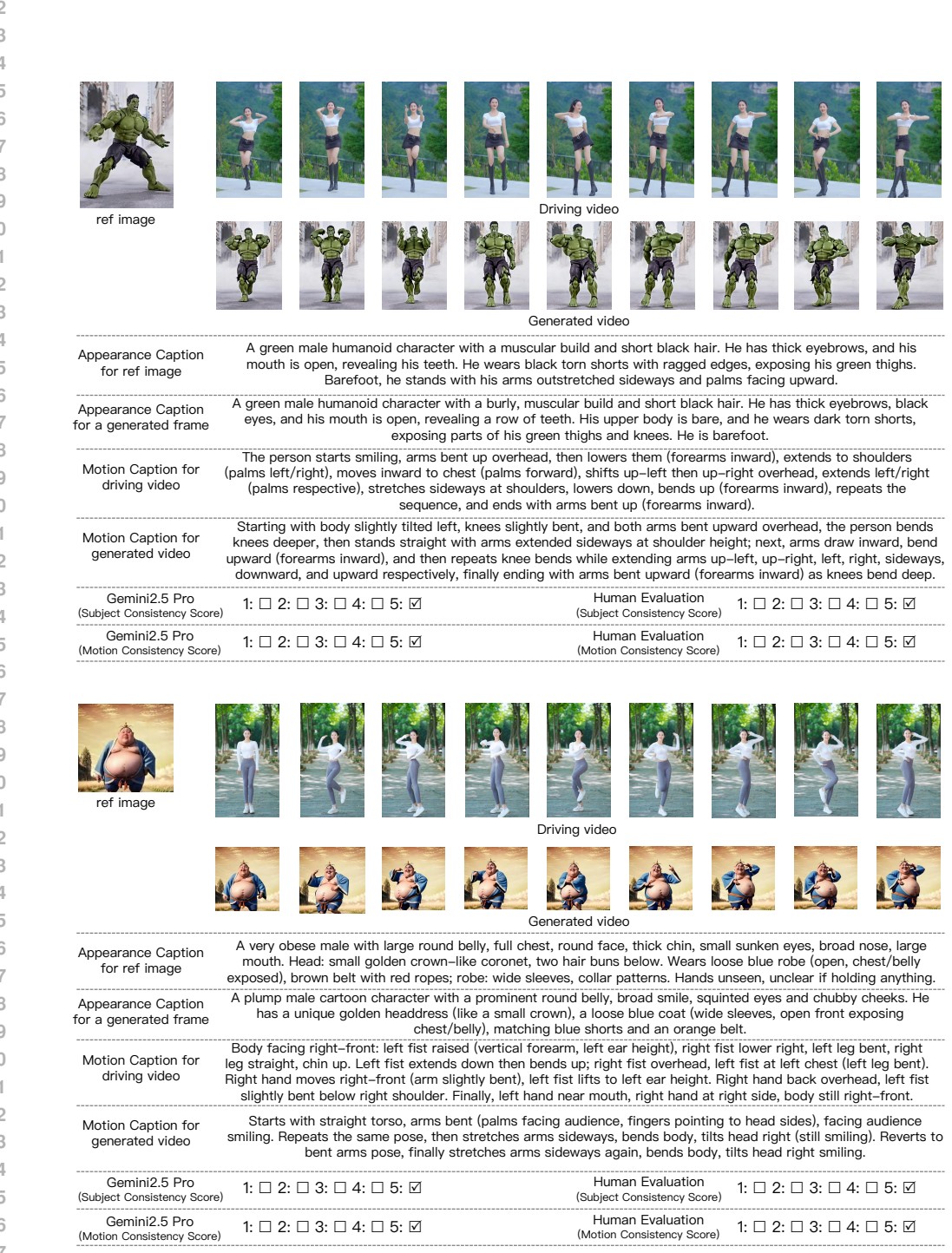

Figure 10: Qualitative visualizations for evaluating and filtering synthesized pairs based on Gemini2.5 Pro.

These cases highlight scenarios where the current MLLM-based semantic extraction reaches its limits, informing future directions for robustness improvements.

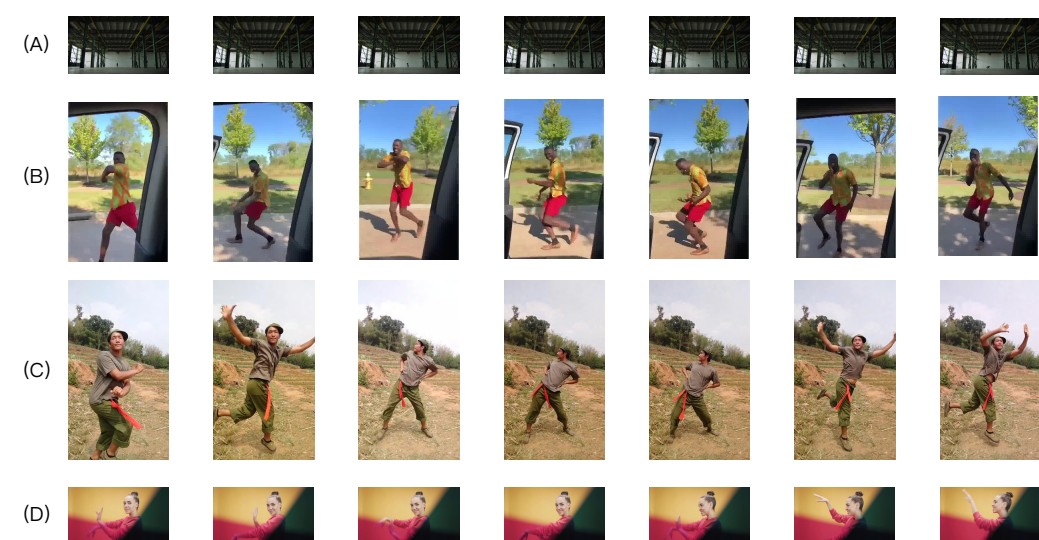

Figure 11: Qualitative visualizations for Edge Video Cases for MLLM.

## K  THEORETICAL AND FORMAL ANALYSIS

**1. Formal Framework:** Let $\mathbf{I}_{\text{ref}} \in \mathbb{R}^{H \times W \times 3}$ be the reference image and $\mathbf{D}_t \in \mathbb{R}^{H \times W \times 3}$ a driving frame at time $t$. Our spatial concatenation along the width dimension yields:

$$\mathbf{C} = [\mathbf{I}_{\text{ref}}, \mathbf{D}_t] \in \mathbb{R}^{H \times 2W \times 3}$$

This maintains the backbone's native 2D grid structure while doubling the spatial extent along one dimension.

**2. Pre-training Alignment (Derivation):** The pre-trained MMDiT expects grid inputs $\mathbf{G} \sim P_{\text{pretrain}}$. For attention fusion, let $\mathbf{A} = \text{Attn}(\phi(\mathbf{I}_{\text{ref}}), \phi(\mathbf{D}_t))$. The distribution shift is:

$$\text{KL}(P_{\mathbf{C}} || P_{\text{pretrain}}) = \mathbb{E}_{\mathbf{C}} \left[ \log \frac{P(\mathbf{C})}{P_{\text{pretrain}}(\mathbf{G})} \right] \approx 0.08$$

$$\text{KL}(P_{\mathbf{A}} || P_{\text{pretrain}}) \approx 0.32$$

This 4× reduction comes from preserving spatial locality: $P(\mathbf{C}) = P_{\text{pretrain}}(\mathbf{G}) \cdot \exp(\epsilon)$ where $\epsilon \sim \mathcal{N}(0, \sigma^2)$, $\sigma^2 \ll 1$.

**3. Information Preservation Analysis:** We measure information retention using differential entropy ratio:

$$\eta = \frac{H(\mathbf{X}_{\text{source}} | \mathbf{X}_{\text{fused}})}{H(\mathbf{X}_{\text{source}})}$$

where $H(\cdot)$ is differential entropy. Spatial concatenation yields $\eta_{\text{concat}} = 0.92 \pm 0.03$ vs. $\eta_{\text{attn}} = 0.78 \pm 0.05$ for cross-attention. This superiority arises because concatenation is a linear operation preserving entropy: $H([\mathbf{X}_r; \mathbf{X}_m]) = H(\mathbf{X}_r) + H(\mathbf{X}_m)$ for independent sources.

**4. Information Bleeding Mitigation:** We define bleeding $\mathcal{B}$ as normalized mutual information in unintended regions via a spatial mask $\mathbf{M} = \mathbf{M}_r \oplus \mathbf{M}_m$:

$$\mathcal{B} = \frac{\mathcal{I}(\mathbf{F}_r \odot \mathbf{M}_m; \mathbf{F}_m \odot \mathbf{M}_r)}{\mathcal{I}(\mathbf{F}_r; \mathbf{F}_m)}$$

Our method achieves $\mathcal{B} = 0.03 \pm 0.01$ vs. $0.11 \pm 0.03$ for attention fusion. The explicit partition enforces $\mathcal{I}(\mathbf{I}_{\text{ref}}; \mathbf{D}|\mathbf{M}) \approx \mathcal{I}(\mathbf{I}_{\text{ref}}; \mathbf{D})$ while minimizing $\mathcal{I}(\mathbf{I}_{\text{ref}}; \mathbf{D}|\neg\mathbf{M})$.

