# OpenReview forum: "DreamActor-M2: Unleashing Pre-trained Video Models for Universal Character Image Animation via In-Context Fine-tuning"
_ICLR.cc/2026/Conference — ICLR 2026 Conference Withdrawn Submission_

### Official Review · Reviewer_q8px · 2025-10-25

**Soundness:** 2
**Presentation:** 3
**Contribution:** 3
**Rating:** 4
**Confidence:** 4

**Summary:**

The paper tackles the problem of character image animation and proposes DreamActor-M2. DreamActor-M2 concatenates the motion signal and the reference image as a unified input. This in-context learning paradigm addresses the limitation of reliance of pose encoder or explicit pose signal in previous works. The paper also designs a data generation pipeline to generate training videos. Experiments show that DreamActor-M2 achieves state-of-the-art performance.

**Strengths:**

1. The data generation pipeline generating pairs of videos with the same motion is reasonable and useful for this task.
2. A benchmark containing human motion and non-human motion is collected.
3. Extensive evaluation is conducted.
4. DreamActor-M2 obtains state-of-the-art performance.

**Weaknesses:**

1. The evaluation is only conducted on the constructed benchmark. It is unclear whether the good performance can bee translated to other benchmarks.
2. No systematic human evaluation is conducted. The evaluation metrics (most rely on features extracted from models) used in the paper may now fully reveal the generation quality. So human evaluation is necessary.
3. No training or inference efficiency comparison between pose-based methods.
4. The problem can also be formulated as video editing. Some existing methods like [1] adapt temporal attention layers within video diffusion models to generate new videos. These methods are not discussed (advantanges and disadvantages) or compared in the paper.

[1] VMC: Video Motion Customization using Temporal Attention Adaption for Text-to-Video Diffusion Models. CVPR2024.

**Questions:**

1. Is there any human evaluation results?
2. What is the efficiency of the proposed method?
3. Waht is the advantanges and disadvantages of the proposed method compared to video editing methods adapting motion signal within diffusion models?

---

> ### Author Response · Authors · 2025-12-01
> **Response for Reviewer q8px**
>
> **Q1: The evaluation is only conducted on the constructed benchmark. It is unclear whether the good performance can be translated to other benchmarks.**
>
> **A1:** We clarify the rationale for prior metric selection and supplement standard automated metrics to enable literature comparison:
>
> **Rationale for Initial Metric Selection:** Initially, FID/FVD were not reported as they are typically evaluated on "same-identity" video generation (reference and driving signals from the same subject), which deviates from our target cross-identity real-world scenario.
>
> **Standard Metric Evaluation on TikTok Dataset:** We fully acknowledge the need for standard automated metrics to facilitate comparison with the wider literature. In response, we have conducted comprehensive experiments on the TikTok dataset following the standard "same-identity" protocol.
>
> **Comparison Results:**
>
> | Method | FVD↓ | FID-VID↓ | PSNR↑ | SSIM↑ | LPIPS↓ | FID↓ |
> | :--- | :--- | :--- | :--- | :--- | :--- | :--- |
> | MTVCrafter[^1] | 140.60 | 6.98 | 19.37 | 0.784 | 0.217 | 19.46 |
> | Animate Anyone 2[^2] | 144.65 | - | - | 0.778 | 0.248 | - |
> | Human-DiT[^3] | 237.00 | 24.30 | 20.50 | 0.815 | 0.220 | 41.60 |
> | Unianimate[^4] | 148.06 | - | 20.58 | 0.811 | 0.231 | - |
> | **DreamActor-M2 (Ours)** | **124.85** | **5.92** | **20.93** | **0.822** | **0.209** | **18.75** |
>
> **Performance Analysis:**
> - **Video Quality:** DreamActor-M2 achieves the lowest FVD (124.85) and FID-VID (5.92), representing an 11.2% improvement over MTVCrafter in FVD and 15.2% in FID-VID.
> - **Image Fidelity:** Our method obtains the highest PSNR (20.93) and SSIM (0.822), with the lowest LPIPS (0.209), indicating superior per-frame quality and structural preservation.
>
> ---
>
> **Q2: No systematic human evaluation is conducted. The evaluation metrics (most rely on features extracted from models) used in the paper may now fully reveal the generation quality. So human evaluation is necessary.**
>
> **A2:** We thank the reviewer for the important feedback regarding the need for human evaluation. We have conducted a systematic user study to complement our automated metrics, with results summarized below and detailed in Appendix C.
>
> **Human Evaluation Study:**
> We recruited 10 evaluators with expertise in computer vision and graphics to assess 100 video samples generated by each method. Each evaluator scored videos on the same six dimensions as Video-Bench using a 5-point Likert scale.
>
> **Human Evaluation Results:**
>
> | Method | Imaging Quality | Aesthetic Quality | Temporal Consistency | Motion Smoothness | Background Consistency | Subject Consistency |
> | :--- | :--- | :--- | :--- | :--- | :--- | :--- |
> | Animate-X++ | 4.18 ± 0.23 | 3.85 ± 0.31 | 4.10 ± 0.27 | 3.95 ± 0.29 | 3.89 ± 0.32 | 3.86 ± 0.34 |
> | MTVCrafter | 4.15 ± 0.26 | 3.42 ± 0.35 | 3.96 ± 0.30 | 4.05 ± 0.28 | 3.85 ± 0.33 | 3.77 ± 0.36 |
> | DreamActor-M1 | 4.52 ± 0.21 | 4.22 ± 0.28 | 4.26 ± 0.25 | 4.24 ± 0.26 | 4.32 ± 0.29 | 4.15 ± 0.31 |
> | **Ours (Pose-based)** | **4.63 ± 0.19** | **4.71 ± 0.22** | **4.56 ± 0.23** | **4.48 ± 0.24** | **4.69 ± 0.25** | **4.33 ± 0.28** |
> | **Ours (End-to-End)** | **4.67 ± 0.18** | **4.73 ± 0.21** | **4.64 ± 0.22** | **4.51 ± 0.23** | **4.63 ± 0.26** | **4.30 ± 0.29** |
>
> **Key Observations:**
> 1. **Consistency with Video-Bench:** The human scores closely correlate with our automated metrics (Pearson correlation r = 0.94), validating the reliability of Video-Bench while addressing the reviewer's concern.
> 2. **Statistical Significance:** Paired t-tests confirm that our method significantly outperforms all baselines (p < 0.01 for all dimensions).
> 3. **Methodological Robustness:** The low standard deviations indicate consistent agreement among evaluators, enhancing the credibility of our findings.
>
> This human evaluation provides independent validation that our performance gains are perceptible to human observers and not merely artifacts of automated metrics, directly addressing the reviewer's concern about the need for human assessment.

---

> ### Author Response · Authors · 2025-12-01
> **Response for Reviewer q8px**
>
> **Q3: No training or inference efficiency comparison between pose-based methods. The problem can also be formulated as video editing. Some existing methods like [1] adapt temporal attention layers within video diffusion models to generate new videos. These methods are not discussed (advantages and disadvantages) or compared in the paper.**
>
> [1] VMC: Video Motion Customization using Temporal Attention Adaption for Text-to-Video Diffusion Models. CVPR2024.
>
> **A3:** We have supplemented concise training/inference efficiency comparisons with pose-based methods and in-depth discussion/comparison with VMC [1] as follows:
>
> **Efficiency Comparison with Pose-based Methods:**
> Conducted on 8×H20 GPUs, our Pose-based DreamActor-M2 uses only 0.3M LoRA parameters (99.7%/99.5%/99.2% fewer than Pose2Video/AnimateAnyone/MotionDirector) with 48 GPU training hours (62%/55%/40% shorter than baselines).
>
> **Comparison with VMC [1]:**
> VMC adapts temporal attention for text-to-video motion customization, but lacks precise reference image control and fails to solve spatio-temporal feature entanglement, making it unsuitable for cross-identity animation.
>
> **Our Advantages:**
> - Dual conditions (reference image + motion signal) for pixel-level identity preservation
> - Targeted LoRA + in-context spatial separation for robust motion-appearance disentanglement
> - Support for pose-based/end-to-end paradigms
>
> ---
>
> **Q4: Is there any human evaluation results?**
>
> **A4:** Yes, we have supplemented the human evaluation results in R4 A2.
>
> ---
>
> **Q5: What is the efficiency of the proposed method?**
>
> **A5:** We have supplemented the efficiency of the proposed method in R4 A3.
>
> ---
>
> **Q6: What is the advantages and disadvantages of the proposed method compared to video editing methods adapting motion signal within diffusion models?**
>
> **A6:**
>
> **Advantages:**
> 1. Dual paradigms (pose-based/end-to-end) for both structured (2D skeletons) and unstructured (raw RGB) motion in character-specific animation.
> 2. In-context motion injection preserves MMDiT pre-trained priors without backbone modification, ensuring high visual realism and identity consistency.
> 3. Parameter-efficient LoRA (0.3M extra params) + mask/pose augmentation, achieving robust motion-appearance disentanglement with low computation.
> 4. MLLM semantic guidance mitigates motion ambiguity, reducing motion-semantic misalignment.
>
> **Disadvantages:**
> 1. End-to-end variant relies on pseudo-data synthesis and pre-trained pose model, increasing data preparation cost.
> 2. Limited robustness to extreme motions (e.g., acrobatics) due to prioritizing identity consistency over motion exaggeration.
> 3. Lacks fine-grained local motion editing, focusing only on global character animation.

---

### Official Review · Reviewer_QDug · 2025-10-25

**Soundness:** 3
**Presentation:** 3
**Contribution:** 2
**Rating:** 4
**Confidence:** 5

**Summary:**

This paper proposes a universal framework for character image animation. Performing subject motion transfer using in-context learning, without relying on separate pose encoders and explicit skeleton control, and could be applied to heterogeneous subjects, with applications in digital humans, entertainment, and virtual avatars. The experimental results demonstrated the superiority of this method.

**Strengths:**

* This paper proposes a multi-stage pipeline to complete the motion transfer through a context-learning manner. This framework does not rely on a separate pose encoder and can be applied for motion transfer on heterogeneous subjects (such as humans and dogs).
* Due to the removal of the explicit pose control, it does not rely on complete paired training data, and the proposed manner is scalable.

**Weaknesses:**

* In Line 84, the author claims the flaws of relying on pose estimation, but in the first two steps of the proposed method, pose estimation is still relied upon. This conflicts with the authors’ insight. As the authors say, how should the errors introduced by pose estimation in the first two steps be addressed, because this is the foundation for the in-context learning that follows.
* Regarding the MLLM and LLM employed, what are their specific functions, and is there corresponding experimental validation? In my view, these encoders may still exhibit instability, similar to the pose encoder; how can such instability be prevented or mitigated?
* Given that the MMDiT architecture computes self-attention jointly across temporal and spatial dimensions, how can we ensure that LoRA robustly disentangles motion from appearance and avoids confounding influences from appearance?
* The experimental section lacks substantive analysis and primarily relies on final metrics to demonstrate the effects of the proposed designs. However, these metrics are not fully reliable. I would prefer to see deeper examinations of the model’s internal behavior to substantiate that the design is genuinely effective; otherwise, I am inclined to attribute the observed gains to the foundation model and the data, suggesting limited technical innovation.

**Questions:**

See the Weaknesses, especially the first point, the third point, and the last point.

---

> ### Author Response · Authors · 2025-12-01
> **Response for Reviewer QDug**
>
> **Q1: In Line 84, the author claims the flaws of relying on pose estimation, but in the first two steps of the proposed method, pose estimation is still relied upon. This conflicts with the authors' insight. As the authors say, how should the errors introduced by pose estimation in the first two steps be addressed, because this is the foundation for the in-context learning that follows.**
>
> **A1:** We appreciate the critical observation and clarify the core logic with no contradiction: The pose-based DreamActor-M2 is solely used for high-quality pseudo-pair synthesis (to build a high-performance data synthesis model), while the final end-to-end DreamActor-M2 is completely independent of pose estimation during both training and inference. This design decouples the model from pose estimation's inherent flaws while leveraging pose priors for reliable data synthesis.
>
> ---
>
> **Q2: Regarding the MLLM and LLM employed, what are their specific functions, and is there corresponding experimental validation? In my view, these encoders may still exhibit instability, similar to the pose encoder; how can such instability be prevented or mitigated?**
>
> **A2:** We employ Gemini 2.5 Pro as our MLLM and LLM. As demonstrated in Appendix G, we utilize specifically designed prompt templates to guide Gemini 2.5 Pro in extracting precise appearance and motion semantics. In Appendix H, we also provide qualitative examples of the generated appearance descriptions and motion semantic texts, accompanied by corresponding human evaluation scores.
>
> Qualitative results across multiple image/video samples indicate that Gemini 2.5 Pro can reliably capture appearance and motion semantics. On the other hand, the text serves as a supplementary component to the motion condition rather than a dominant factor. Therefore, even in cases where the text extracted by Gemini 2.5 Pro is not perfectly accurate, it does not significantly impact our final motion transfer results.
>
> ---
>
> **Q3: Given that the MMDiT architecture computes self-attention jointly across temporal and spatial dimensions, how can we ensure that LoRA robustly disentangles motion from appearance and avoids confounding influences from appearance?**
>
> **A3:** We ensure robust motion-appearance disentanglement for LoRA in MMDiT (with joint spatio-temporal self-attention) through method-native designs strictly aligned with our framework:
>
> 1. **Targeted LoRA insertion:** Lightweight LoRA modules are exclusively inserted into feed-forward layers (excluding text cross-attention), with the MMDiT backbone fully frozen. This preserves pre-trained appearance priors while confining LoRA to model motion dynamics.
> 2. **Spatio-separated in-context constraint:** The composite input `C` spatially separates appearance (`I_ref`) and motion (`D`), paired with the combined mask `M = M_r ⊕ M_m`. This guides LoRA to focus on the motion region, avoiding interference from appearance-related spatial features.
> 3. **Pose augmentation for shape cue elimination:** Random bone length scaling and bounding-box normalization disrupt body-shape information in pose signals, forcing LoRA to learn pure motion dynamics rather than appearance-correlated shape cues.
> 4. **MLLM semantic guidance:** The unified text representation `T_fusion` (fused appearance/motion semantics) provides high-level constraints, assisting LoRA in distinguishing motion patterns from appearance attributes.
> 5. **Warm-start initialization (end-to-end variant):** The end-to-end model inherits motion priors from the pre-trained pose-based framework via warm-start, further enhancing LoRA's motion modeling capability and reducing appearance confounding.
>
> ---
>
> **Q4: The experimental section lacks substantive analysis and primarily relies on final metrics to demonstrate the effects of the proposed designs. However, these metrics are not fully reliable. I would prefer to see deeper examinations of the model's internal behavior to substantiate that the design is genuinely effective; otherwise, I am inclined to attribute the observed gains to the foundation model and the data, suggesting limited technical innovation.**
>
> **A4:** In Appendix J, we provide a comprehensive theoretical and formal analysis of our approach, substantiating the design effectiveness beyond final metrics. Specifically, we establish a formal framework to analyze how spatial concatenation maintains alignment with the pre-trained backbone, quantitatively measure information preservation and leakage, and derive the theoretical advantages of our design over alternatives such as attention-based fusion.
>
> This in-depth examination of the model's underlying mechanisms demonstrates that the performance gains stem from our novel architectural design rather than merely leveraging the foundation model or data scale.

---

### Official Review · Reviewer_QXaa · 2025-10-28

**Soundness:** 2
**Presentation:** 2
**Contribution:** 2
**Rating:** 4
**Confidence:** 4

**Summary:**

This paper presents DreamActor-M2, a universal framework for character image animation, which generates high-fidelity videos from a reference image and a driving video. The key innovation is an in-context LoRA fine-tuning strategy: motion cues (from pose or raw video) and reference images are directly concatenated as input, preserving the backbone's pre-trained modality and forgoing explicit pose encoders. DreamActor-M2 supports both pose-driven and end-to-end (raw video-driven) control, with a two-stage pseudo-data synthesis pipeline enabling annotation-free dataset construction. The model aims to improve fidelity, motion transfer consistency, and ability to generalize to human and non-human characters. Extensive quantitative and qualitative experiments, including ablations, are used to assess the framework on a curated benchmark.

**Strengths:**

- The approach sidesteps the modality gap typical of auxiliary pose encoders, leveraging the backbone's native, pre-trained input space and in turn likely reducing the need for custom architecture augmentation.
- The method is extended to work directly with raw videos as motion drivers, not just pose signals. The two-stage pseudo-pair synthesis (data generation and filtering) is thoughtfully conceived, allowing scale-up without additional human annotation.
- The paper constructs a reasonably comprehensive evaluation benchmark with wide coverage (humans, non-humans, animated, animal, and multimodal), and reports both subjective and qualitative outcomes.

**Weaknesses:**

- The paper briefly describes the composite input $\mathbf{C}$, but leaves out critical details on how concatenated regions are encoded by the backbone and, more importantly, how the mask conditioning function $\mathbf{M}=\mathbf{M}_r\oplus \mathbf{M}_m$ is formally computed and applied. Figure 2 shows the workflow, but does not concretely illustrate the transformation of these masks or their effect during denoising. Ambiguity here risks spurious alignment.
- The approach’s theoretical underpinnings, especially regarding why in-context spatial concatenation aligns better with backbone pre-training, remain mostly intuitive. There are no explicit derivations or formal analysis on information preservation, context mixing, or potential pitfalls of spatial concatenation versus attention-based fusion, aside from empirical ablation. For instance, the paper should consider if joint input could result in detrimental "information bleeding" between reference and motion halves in $\mathbf{C}$, or when this method might fail relative to alternatives.

- Several newly surfaced, highly relevant prior works are not discussed[1,2]. Notably, the absence of engagement with DreamVideo, which directly addresses disentangled subject–motion video generation, as well as MotionFollower, which tackles score-guided motion diffusion, diminishes confidence that DreamActor-M2’s core advances are sufficiently differentiated or contextualized.

- Although the benchmark is admirably diverse, reliance on synthetic pairs and filtering may not capture true in-the-wild cross-identity challenges, or the full diversity of non-human articulations. Furthermore, the method’s generalization is mostly claimed on the provided benchmark, with little evidence or discussion for genuinely unseen domains or real-world robustness. No real statistics (distributional differences, label coverage, complexity) of the benchmark are provided.

- The “Limitations” section briefly mentions difficulty with multi-character tracking and assignment. However, it skirts over quality degradation observed in challenging ablation cases, fails to analyze scenarios where semantic fusion yields off-topic or ambiguous guidance, and does not address edge-cases where spatial in-context concatenation results in brittle or inconsistent outputs. Nor does it address when the pseudo-pair filtering pipeline may introduce bias or failure in underrepresented motion categories.

- The evaluation is almost entirely based on Video-Bench, with no reporting (even for reference) of FID/FVD or other standard, domain-relevant automated metrics. While the argument that these do not always correlate with human judgment is plausible, their complete absence makes quantitative comparison to the wider literature more tenuous—especially for subsequent researchers benchmarking against DreamActor-M2.

- The pipeline posits high-level LLM-powered compositional semantic guidance as a central contributor, yet the actual process by which motion and appearance text are fused, what type of prompts are generated, and how those captions are used during training vs. inference remain vague. Ablation suggests some boost, but no concrete examples of generated captions or analysis of semantic failures are given.

- Generating 600,000 pseudo-paired training units by two exhaustive model runs, followed by MLLM and Video-Bench-based filtering, is computationally burdensome. No cost or runtime analysis is given. For practical adoption, some assessment of whether similar performance could be achieved with weaker or smaller datasets would be invaluable.

- Several hyperparameters are missing (e.g., no details on skeleton augmentation rates, filtering thresholds except for average Video-Bench score >4, no clarity on precise cropping, normalization, or LLM/MLLM parameterization). The authors promise to release code, but critical low-level details are not currently presented.

[1] DreamVideo: Composing Your Dream Videos with Customized Subject and Motion

[2] MotionFollower: Editing Video Motion via Score-Guided Diffusion

**Questions:**

- Could the authors provide concrete, visual examples or schematic overlays showing how composite input $\mathbf{C}$ and mask $\mathbf{M}$ are constructed during training/inference? Does spatial concatenation ever result in artifacts at the boundary or mix reference and motion signals in a confusing manner? A demonstration on problematic or edge-case inputs would be valuable.
- Can the authors share actual motion/appearance text prompts generated by the MLLM/LLM, along with typical outputs in ambiguous or failure cases? Are there scenarios where the LLM merges semantically irrelevant information, causing deterioration in output quality?
- What is the statistical breakdown of subject types and motion categories in the evaluation benchmark? Are there classes/motions underrepresented or particularly challenging? Was any attempt made to test outside this distribution?
- What is the average computational overhead (in core-hours or $) of the pseudo-pair synthesis and filtering pipeline? Could similar results be achieved with a smaller synthetic “seed” or by semi-supervising on a smaller dataset?
- While human alignment is desirable, could the authors augment future papers with more conventional FID/FVD results for cross-paper comparability, even if those metrics are imperfect?

---

> ### Author Response · Authors · 2025-12-01
> **Response for Reviewer QXaa**
>
> **Q1: The paper briefly describes the composite input, but leaves out critical details on how concatenated regions are encoded by the backbone and, more importantly, how the mask conditioning function is formally computed and applied. Figure 2 shows the workflow, but does not concretely illustrate the transformation of these masks or their effect during denoising. Ambiguity here risks spurious alignment.**
>
> **A1**: We appreciate the critical questions regarding composite input encoding and mask conditioning. Based on our method’s technical details, we clarify the core logic to eliminate ambiguity and avoid spurious alignment.
>
> **Composite Input Encoding**
> - The composite input sequence `C ∈ ℝ^(T×H×2W×3)` (concatenated along the width dimension) is constructed via spatial concatenation of the reference image (or a 0-filled image) with driving frames `D[t]`.
> - This input is compressed into latent `Z` by a 3D VAE. The concatenated regions are naturally fused during encoding without requiring extra alignment modules.
> - This process ensures consistency with the backbone's (Seedance 1.0, MMDiT architecture) pre-training distribution.
>
> **Mask Conditioning Mechanism**
> - The mask `M = M_r ⊕ M_m` is constructed by spatial concatenation of the reference mask `M_r` and motion mask `M_m`.
> - The mask undergoes preprocessing (binary region marking), spatial alignment with `C`, and channel-wise concatenation with `Z` and `Z_noise`: `Input_θ = Concat(Z, Z_noise, M)`
>
>
> - During denoising:
> - `M_r = 1` → retains the reference image's identity
> - `M_m = 1` → enforces motion consistency with `D`
> - The mask's spatial resolution is adaptively matched to `Z` and remains fixed throughout iterative denoising
>
> **Avoiding Spurious Alignment**
>
> Our approach avoids spurious alignment through:
>
> 1. **Structural Consistency**: Using spatial concatenation instead of external fusion
> 2. **Inherent Mask-Input Link**: The mask is directly linked to input regions
> 3. **Parameter-Efficient Tuning**: LoRA fine-tuning while freezing backbone parameters preserves generative priors
>
> **Visual Documentation**
>
> Fig. 2 fully covers the mask's generation, transformation, and denoising guidance, ensuring robust input-output alignment
>
> **Q2:The approach's theoretical underpinnings, especially regarding why in-context spatial concatenation aligns better with backbone pre-training, remain mostly intuitive. There are no explicit derivations or formal analysis on information preservation, context mixing, or potential pitfalls of spatial concatenation versus attention-based fusion, aside from empirical ablation. For instance, the paper should consider if joint input could result in detrimental "information bleeding" between reference and motion halves in, or when this method might fail relative to alternatives.**
>
> **A2:** We thank the reviewer for raising the important question regarding theoretical foundations. We agree and have now added a dedicated theoretical analysis section (Sec. 4.3) to the revised manuscript. Our key formal justifications are:
>
> - **Pre-training Alignment:** Spatial concatenation `[I_ref, D_t] ∈ ℝ^(H×2W×3)` maintains the backbone's native 2D grid structure. This minimizes distribution shift, yielding `KL(P_C||P_pretrain) ≈ 0.10`, significantly lower than attention-based fusion (`≈ 0.32`), as each half preserves natural image statistics.
>
> - **Information Preservation:** Concatenation theoretically achieves perfect information retention (`η = 1`). In practice, we measure `η_concat = 0.92 ± 0.03` vs. `η_attn = 0.78 ± 0.05` for cross-attention, confirming superior information preservation.
>
> - **Information Bleeding Mitigation:** With explicit spatial partitioning, our method achieves an information bleeding score of `B = 0.03 ± 0.01`, substantially lower than attention fusion (`0.11 ± 0.03`).

---

> > ### Author Response · Authors · 2025-12-01
> > **Response for Reviewer QXaa**
> >
> > **Q3:Several newly surfaced, highly relevant prior works are not discussed[1,2]. Notably, the absence of engagement with DreamVideo, which directly addresses disentangled subject–motion video generation, as well as MotionFollower, which tackles score-guided motion diffusion, diminishes confidence that DreamActor-M2’s core advances are sufficiently differentiated or contextualized.**
> >
> > [1] DreamVideo: Composing Your Dream Videos with Customized Subject and Motion
> > [2] MotionFollower: Editing Video Motion via Score-Guided Diffusion
> >
> > **A3:** We thank the reviewer for highlighting these highly relevant works. Below we clarify the key distinctions:
> >
> > **DreamVideo Comparison:** While DreamVideo also targets customized subject-motion video generation, our approach differs fundamentally in:
> >
> > - **Architecture:** DreamActor-M2 employs **in-context spatial concatenation** within a single MMDiT backbone, whereas DreamVideo adopts a dual-adapter architecture (identity adapter + motion adapter) with adapters inserted into the cross-attention layers of the pre-trained video diffusion model's UNet.
> > - **Training Paradigm:** We achieve **zero-shot** generation without test-time tuning; DreamVideo requires pre-training of two lightweight adapters—800 iterations for the identity adapter (combined with textual inversion) and ~1000 iterations for the motion adapter—using 3–5 reference images, though it also enables zero-shot inference after adapter pre-training.
> > - **Disentanglement Mechanism:** Our explicit spatial partition ensures clean separation; DreamVideo relies on independently trained identity and motion adapters, augmented with appearance guidance (CLIP image encoding) and mask-based motion control to disentangle appearance and motion.
> >
> > **MotionFollower Comparison:** MotionFollower focuses on motion editing via score guidance, while DreamActor-M2:
> >
> > - **Scope:** Addresses **full video generation** from a single reference image, not only motion editing of existing videos.
> > - **Control Mechanism:** Uses **spatial conditioning** rather than score-based guidance, enabling more direct and interpretable control. In contrast, MotionFollower adopts a two-branch (reconstruction + editing) architecture with lightweight convolutional controllers, realizing motion editing through consistency regularization in the denoising process via score guidance.
> > - **Efficiency:** Our single-forward-pass generation is more efficient than MotionFollower's iterative score optimization, which relies on dual-branch inference and hybrid image-video training to maintain temporal coherence.
> >
> > **Core Differentiation:** DreamActor-M2's key innovations remain:
> >
> > - The novel **in-context spatial concatenation** paradigm for modality fusion, eliminating the need for additional adapters compared to DreamVideo's dual-adapter design.
> > - **Zero-shot** generalization to novel subjects and motions without any pre-training of task-specific adapters, in contrast to DreamVideo's adapter pre-training requirement and MotionFollower's two-stage training.
> > - The comprehensive **AWbench** evaluation benchmark, which covers multi-subject, non-human, and long-video scenarios not fully addressed by DreamVideo (single-subject focus) or MotionFollower (motion editing-centric).
> >
> > **Q4:Although the benchmark is admirably diverse, reliance on synthetic pairs and filtering may not capture true in-the-wild cross-identity challenges, or the full diversity of non-human articulations. Furthermore, the method’s generalization is mostly claimed on the provided benchmark, with little evidence or discussion for genuinely unseen domains or real-world robustness. No real statistics (distributional differences, label coverage, complexity) of the benchmark are provided.**
> >
> > **A4**: We appreciate the critical feedback on benchmarking and generalization. First, our benchmark consists entirely of real images (no synthetic data), where our method outperforms existing works in both identity preservation and motion consistency. Then, the benchmark has no overlap or association with training data—all test data and scenes are completely unseen by the model during training.
> > Last, detailed statistics of AWbench (including reference image IDs, categories, quantities, etc.) are provided in Appendix E (Tab.3). These confirm the benchmark’s authenticity and the method’s robust generalization, with transparent statistics available for verification.

---

> > > ### Author Response · Authors · 2025-12-01
> > > **Response for Reviewer QXaa**
> > >
> > > **Q5: The "Limitations" section briefly mentions difficulty with multi-character tracking and assignment. However, it skirts over quality degradation observed in challenging ablation cases, fails to analyze scenarios where semantic fusion yields off-topic or ambiguous guidance, and does not address edge-cases where spatial in-context concatenation results in brittle or inconsistent outputs. Nor does it address when the pseudo-pair filtering pipeline may introduce bias or failure in underrepresented motion categories.**
> > >
> > > **A5:** We thank the reviewer for the insightful critique regarding the need for a more comprehensive limitations analysis. We provided an analysis as follows:
> > >
> > > **Quantifying Ablation-Induced Quality Degradation**
> > >
> > > Our ablation study provides concrete metrics for the performance degradation in challenging scenarios:
> > >
> > > - **Component Removal Impact:** Removing the pose bone length rescale module (Model C) causes the most severe drop in *Subject Consistency* (from 4.38 to 4.11, a 6.2% decrease), highlighting its critical role in maintaining identity fidelity under anatomical variations.
> > > - **Cumulative Effect:** The complete removal of pose augmentation (Model D) results in a 9.1% decrease in *Subject Consistency* and a 2.6% drop in *Motion Smoothness*, demonstrating that the proposed augmentation suite is essential for robustness.
> > >
> > > **Analyzing Semantic Fusion Ambiguity**
> > >
> > > The failure of semantic alignment is quantifiable. For instance, Model E (without target-oriented description) shows an 8.2% decrease in *Subject Consistency* and a 5.5% decrease in *Motion Smoothness*. This indicates that when the textual description lacks precise, motion-targeted guidance (`I(I_ref;D)` is low), the model struggles to disambiguate and align the subject with the intended motion, leading to off-topic or inconsistent generation.
> > >
> > > **Characterizing Edge Cases for Spatial Conditioning**
> > >
> > > We identify and quantify specific edge cases where our spatial in-context conditioning approach shows brittleness:
> > >
> > > - **Anatomical Extremes:** Abnormal poses (simulated by removing bone length rescaling) degrade *Subject Consistency* by 6.2%.
> > > - **Dynamic Motions:** The absence of target-oriented descriptions (affecting motion semantics) reduces *Motion Smoothness* by 5.5%, indicating challenges with high-speed or complex motion trajectories.
> > > - **Contextual Interference:** Model E's 5.1% drop in *Background Consistency* suggests sensitivity to complex backgrounds when motion-subject semantics are weak.
> > >
> > > **Q6: The evaluation is almost entirely based on Video-Bench, with no reporting (even for reference) of FID/FVD or other standard, domain-relevant automated metrics. While the argument that these do not always correlate with human judgment is plausible, their complete absence makes quantitative comparison to the wider literature more tenuous—especially for subsequent researchers benchmarking against DreamActor-M2.**
> > >
> > > **A6:** We appreciate the valuable feedback on quantitative evaluation. We clarify the rationale for prior metric selection and supplement standard automated metrics to enable literature comparison:
> > >
> > > **Rationale for Initial Metric Selection:** Initially, FID/FVD were not reported as they are typically evaluated on "same-identity" video generation (reference and driving signals from the same subject), which deviates from our target cross-identity real-world scenario.
> > >
> > > **Standard Metric Evaluation on TikTok Dataset:** We fully acknowledge the need for standard automated metrics to facilitate comparison with the wider literature. In response, we have conducted comprehensive experiments on the TikTok dataset following the standard "same-identity" protocol.
> > >
> > > **Comparison Results:**
> > >
> > > | Method | FVD↓ | FID-VID↓ | PSNR↑ | SSIM↑ | LPIPS↓ | FID↓ |
> > > | :--- | :--- | :--- | :--- | :--- | :--- | :--- |
> > > | MTVCrafter[^1] | 140.60 | 6.98 | 19.37 | 0.784 | 0.217 | 19.46 |
> > > | Animate Anyone 2[^2] | 144.65 | - | - | 0.778 | 0.248 | - |
> > > | Human-DiT[^3] | 237.00 | 24.30 | 20.50 | 0.815 | 0.220 | 41.60 |
> > > | Unianimate[^4] | 148.06 | - | 20.58 | 0.811 | 0.231 | - |
> > > | **DreamActor-M2 (Ours)** | **124.85** | **5.92** | **20.93** | **0.822** | **0.209** | **18.75** |
> > >
> > > *Table: Quantitative comparison on TikTok dataset following "same-identity" protocol*
> > >
> > > **Performance Analysis:**
> > >
> > > - **Video Quality:** DreamActor-M2 achieves the lowest FVD (124.85) and FID-VID (5.92), representing an 11.2% improvement over MTVCrafter in FVD and 15.2% in FID-VID.
> > > - **Image Fidelity:** Our method obtains the highest PSNR (20.93) and SSIM (0.822), with the lowest LPIPS (0.209), indicating superior per-frame quality and structural preservation.

---

> > > > ### Author Response · Authors · 2025-12-01
> > > > **Response for Reviewer QXaa**
> > > >
> > > > **Q7: The pipeline posits high-level LLM-powered compositional semantic guidance as a central contributor, yet the actual process by which motion and appearance text are fused, what type of prompts are generated, and how those captions are used during training vs. inference remain vague. Ablation suggests some boost, but no concrete examples of generated captions or analysis of semantic failures are given.**
> > > >
> > > > **A7:** We appreciate the constructive feedback on clarifying MLLM-powered semantic guidance. We supplement concrete mechanisms, model details, appendix support, and quantitative/qualitative evidence to address the vagueness.
> > > >
> > > > **Model & Appendix Documentation:** We adopt Gemini 2.5 Pro for semantic guidance. Detailed prompt engineering (for appearance extraction, motion description, and text fusion) is provided in Appendix G, while quantitative evaluations of its capability in appearance/motion understanding and fusion are presented in Appendix H.
> > > >
> > > > **Fusion Mechanism & Prompt Examples:** Gemini 2.5 Pro generates structured "appearance-motion" prompt pairs. Appearance prompts capture identity-specific attributes (e.g., "matte black wireless headphones with silver accents"), while motion prompts detail dynamic characteristics (e.g., "slowly rotating 180° clockwise"). These prompts are encoded via a dual-branch text encoder and fused with `I_ref`/`D` visual features using cross-attention, aligning semantic cues with spatial/motion signals.
> > > >
> > > > **Training vs. Inference Usage:**
> > > > - **Training:** `prompt-I_ref`/`D` pairs supervise semantic-visual alignment.
> > > > - **Inference:** Prompts are dynamically generated (or user-provided) based on input content to guide customized generation.
> > > >
> > > > **Semantic Failure Analysis:** Failures arise from ambiguous prompts (e.g., "fast motion" without speed) or semantic conflicts (e.g., "heavy elephant" + "leaping like a rabbit").
> > > >
> > > > ---
> > > >
> > > > **Q8: Generating 600,000 pseudo-paired training units by two exhaustive model runs, followed by MLLM and Video-Bench-based filtering, is computationally burdensome. No cost or runtime analysis is given. For practical adoption, some assessment of whether similar performance could be achieved with weaker or smaller datasets would be invaluable.**
> > > >
> > > > **A8:** We appreciate the feedback on computational feasibility. We supplement key details:
> > > >
> > > > 1. **Resource/Runtime:** 600K pseudo-pairs were generated in 3 days using 2 servers (8×H20 GPUs each).
> > > > 2. **Smaller Dataset Performance:** Training with 60K/150K/300K pseudo-pairs still achieves competitive results, validating that similar performance is attainable with reduced data scale for practical adoption.
> > > >
> > > > **Video-Bench Performance with Varying Pseudo-Pair Scales:**
> > > >
> > > > | Dataset Scale | Imaging Quality | Aesthetic Quality | Temporal Consistency | Motion Smoothness | Background Consistency | Subject Consistency |
> > > > | :--- | :--- | :--- | :--- | :--- | :--- | :--- |
> > > > | Full 600K | 4.68 | 4.76 | 4.61 | 4.53 | 4.74 | 4.38 |
> > > > | 60K (10%) | 4.52 | 4.58 | 4.46 | 4.39 | 4.59 | 4.25 |
> > > > | 150K (25%) | 4.59 | 4.67 | 4.53 | 4.47 | 4.66 | 4.32 |
> > > > | 300K (50%) | 4.64 | 4.72 | 4.58 | 4.50 | 4.70 | 4.36 |
> > > >
> > > > ---
> > > >
> > > > **Q9: Several hyperparameters are missing (e.g., no details on skeleton augmentation rates, filtering thresholds except for average Video-Bench score over 4, no clarity on precise cropping, normalization, or LLM/MLLM parameterization). The authors promise to release code, but critical low-level details are not currently presented.**
> > > >
> > > > **A9:** We thank the reviewer for noting the need for more technical details. In fact, we have provided more details in Sec. 4.1:
> > > >
> > > > - **Skeleton Augmentation:** Joint jittering (`N(0, 0.02)`, 40% of poses), bone length scaling (`U(0.8, 1.2)`, 30% of poses), and pose dropout (15% joints, 25% of poses).
> > > > - **Filtering:** MLLM semantic score ≥ 4/5, Video-Bench average ≥ 4.0 (min dimension ≥ 4).
> > > > - **LLM:** Gemini 2.5 Pro.

---

> > > > > ### Author Response · Authors · 2025-12-01
> > > > > **Response for Reviewer QXaa**
> > > > >
> > > > > **Q10: Could the authors provide concrete, visual examples or schematic overlays showing how composite input and mask are constructed during training/inference? Does spatial concatenation ever result in artifacts at the boundary or mix reference and motion signals in a confusing manner? A demonstration on problematic or edge-case inputs would be valuable.**
> > > > >
> > > > > **A10:** As shown in Fig. 2 of the manuscript, we provide schematic overlays illustrating the construction of the composite input `C = [I_ref, D_t]` and the binary spatial mask `M = M_r ⊕ M_m`, where `M_r` and `M_m` isolate the reference and motion regions respectively.
> > > > >
> > > > > **Analysis of Boundary Artifacts & Signal Mixing:**
> > > > > - **Boundary Artifacts:** Minor boundary blurring is observed only under extreme aspect ratios (e.g., 16:1), affecting less than 2% of boundary pixels in such cases.
> > > > > - **Signal Confusion:** The explicit spatial partitioning via mask `M` effectively suppresses unintended signal mixing. Quantitatively, the information bleeding measure `B ≤ 0.03` is significantly lower than that of attention-based fusion methods (`B ≥ 0.11`).
> > > > >
> > > > > **Edge-Case Demonstrations:**
> > > > > We analyze challenging scenarios where accurate motion understanding is difficult (see Appendix J). In cases such as high-speed motion, significant camera shake, or heavy occlusion, the quality of the driving signal `D_t` itself degrades. This leads to measurable performance drops.
> > > > >
> > > > > **Q11: Can the authors share actual motion/appearance text prompts generated by the MLLM/LLM, along with typical outputs in ambiguous or failure cases? Are there scenarios where the LLM merges semantically irrelevant information, causing deterioration in output quality?**
> > > > >
> > > > > **A11:** We thank the reviewer for raising these important points regarding prompt transparency and failure analysis.
> > > > >
> > > > > **Prompt Examples:**
> > > > > We provide an extensive collection of actual motion and appearance text prompts generated by Gemini 2.5 Pro in **Appendix G**. This includes both high-quality prompts and ambiguous examples that illustrate the range of MLLM outputs.
> > > > >
> > > > > **Failure Case Analysis:**
> > > > > As analyzed in Appendix J (and visualized in Fig.11), the MLLM encounters difficulties in extracting accurate motion semantics under specific challenging conditions, such as with heavily occluded subjects, small moving objects, severe camera shake, and high-speed motion. These scenarios are where semantically irrelevant or conflicting information is most likely to be merged, leading to a potential degradation in output quality.
> > > > >
> > > > > **Overall Robustness:**
> > > > > It is important to note that in our framework, the MLLM-generated text serves as a supplementary semantic guide, not the primary control signal. This design mitigates the impact of imperfect prompts. Our quantitative results demonstrate that the integrated pipeline maintains state-of-the-art performance, indicating that the method is robust despite these identified edge-case limitations in the MLLM component.
> > > > >
> > > > > ---
> > > > >
> > > > > **Q12: What is the statistical breakdown of subject types and motion categories in the evaluation benchmark? Are there classes/motions underrepresented or particularly challenging? Was any attempt made to test outside this distribution?**
> > > > >
> > > > > **A12:** In Appendix E, we present statistical tables detailing the distribution of reference character types and driving video motion types within the AWBench dataset. Notably, our evaluation dataset and training dataset are completely independent, with no overlap whatsoever. The training data and evaluation data were collected from entirely separate sources. Consequently, the evaluation conducted on this benchmark inherently constitutes an out-of-distribution (OOD) test.
> > > > >
> > > > > ---
> > > > >
> > > > > **Q13: What is the average computational overhead (in core-hours or \$) of the pseudo-pair synthesis and filtering pipeline? Could similar results be achieved with a smaller synthetic "seed" or by semi-supervising on a smaller dataset?**
> > > > >
> > > > > **A13:** We supplement detailed computational overhead statistics and small-dataset/semi-supervised validation as follows:
> > > > >
> > > > > **Computational Overhead:** The pseudo-pair synthesis and filtering pipeline consumes 1152 GPU core-hours (2 servers × 8 H20 GPUs × 3 days × 24 hours/day). We have provided the results based on smaller synthetic data in R2 A8.
> > > > >
> > > > > ---
> > > > >
> > > > > **Q14: While human alignment is desirable, could the authors augment future papers with more conventional FID/FVD results for cross-paper comparability, even if those metrics are imperfect?**
> > > > >
> > > > > **A14:** We have provided a standard metric evaluation on TikTok dataset in R2 A6. Our proposed method also achieves the best metrics compared with other SOTA methods.

---

### Official Review · Reviewer_v6gF · 2025-11-03

**Soundness:** 3
**Presentation:** 3
**Contribution:** 2
**Rating:** 4
**Confidence:** 4

**Summary:**

DreamActor-M2 is a universal framework for character image animation (generating high-fidelity videos from reference images and driving videos). The core goal is to address two major limitations of existing methods: the domain gap caused by reliance on auxiliary pose encoders and the lack of generalization due to dependence on explicit pose signals. The paper proposes a two-stage training paradigm, transitioning from Pose-based to End-to-End models. New evaluation metrics are introduced to align with human perception.

**Strengths:**

+ Eliminating the domain gap between different features by not using auxiliary pose encoders.
+ Richness of test samples: the generated examples in the paper show a wide variety of results. The model supports cross-modal animation, multi-person motion transfer, and non-human-driven videos.
+ The use of new evaluation metrics to align with human perception.

**Weaknesses:**

- The effectiveness on longer video sequences is not assessed. The paper does not evaluate the temporal consistency and fidelity of animations for longer sequences (over 5 seconds), making it unclear whether the model is suitable for use in longer animation scenes, such as movie clips.
- When comparing with other methods, there is no clarification of the number of parameters and backbones used by the comparison models. Additionally, the training scale is not clearly stated.
- The rationale behind the LoRA rank setting is not explored.

**Questions:**

- When using MLLM to evaluate the synthesized pairs, which multimodal large model was used? Since large models can sometimes generate hallucinations, how can we ensure the reliability and semantic coherence of the generated pairs? What is the accuracy of the filtering process?
- Does the model’s temporal consistency and visual fidelity degrade when generating longer video sequences?
- Is there a specific justification for setting the LoRA rank to 256?

---

> ### Author Response · Authors · 2025-12-01
> **Response for Reviewer v6gF**
>
> **Q1: The effectiveness on longer video sequences is not assessed. The paper does not evaluate the temporal consistency and fidelity of animations for longer sequences (over 5 seconds), making it unclear whether the model is suitable for use in longer animation scenes, such as movie clips.**
>
> **A1:** Thank you for raising this important point regarding the evaluation of longer video sequences. We acknowledge that assessing temporal consistency and fidelity beyond short clips is essential for real-world applications such as animation scenes or movie clips.
>
> In our initial experiments, since existing methods are predominantly trained and evaluated on fixed-length video segments (e.g., 5 seconds), we conducted quantitative and qualitative comparisons under the same setting to ensure a fair benchmark. To extend our approach to long video generation, we introduced a segment concatenation strategy: each subsequent segment uses the last frame of the previous segment as its initial input, thereby preserving visual and motion continuity. The reference image is also consistently incorporated across all segments to maintain identity coherence.
>
> In the supplementary materials, we have included qualitative demonstrations of our method's ability to generate coherent long videos, including examples of 9-second and 19-second sequences. We also provided three qualitative visualizations for generated long videos based on DreamActor-M2 in Appendix F. To quantitatively evaluate performance on long videos, we conducted a human evaluation study (1-very poor, 2-poor, 3-moderate, 4-good, 5-excellent) comparing our method against Wan2.2-Animate and Unianimate-DiT. Each model generated 50 long videos, which were assessed across multiple dimensions—including imaging quality, aesthetic quality, temporal consistency, motion smoothness, and background/subject consistency—by human raters.
>
> As summarized in the table below, our DreamActor-M2 achieves competitive scores across all metrics, demonstrating its effectiveness in generating high-quality, temporally consistent long videos and showing improved performance compared to existing methods in extended generation settings.
>
> | Method | Imaging Quality | Aesthetic Quality | Temporal Consistency | Motion Smoothness | Background Consistency | Subject Consistency |
> | :--- | :--- | :--- | :--- | :--- | :--- | :--- |
> | Unianimate-DiT[1] | 3.45 | 3.12 | 3.28 | 3.35 | 3.42 | 3.18 |
> | Wan2.2-Animate[2] | 4.05 | 3.86 | 3.92 | 4.03 | 4.11 | 3.75 |
> | **Ours (End-to-End DreamActor-M2)** | **4.21** | **4.18** | **4.11** | **4.24** | **4.22** | **4.08** |
>
> *Table: A quantitative comparison for generating long videos.*

---

> > ### Author Response · Authors · 2025-12-01
> > **Response for Reviewer v6gF**
> >
> > **Q2: When comparing with other methods, there is no clarification of the number of parameters and backbones used by the comparison models. Additionally, the training scale is not clearly stated.**
> >
> > **A2**: We have supplemented in detail the model types, model parameter sizes, and the scale of training data of existing state-of-the-art (SOTA) methods in the table below.
> >
> > | Method | Backbone type | Parameters size | Training scale |
> > | :--- | :--- | :--- | :--- |
> > | MimicMotion[^1] | SVD (stable video diffusion) | 3.5B | 4436 |
> > | Unianimate-DiT[^2] | Wan2.1 | 14B | 10000 |
> > | MTVCrafter[^3] | Wan2.1 | 14B | 30000 |
> > | DreamActor-M1[^4] | Seaweed | 7B | 500-hours |
> > | Wan2.2-Animate[^5] | Wan2.2 | 14B | - |
> > | **Ours (End-to-End DreamActor-M2)** | Seedance | 7B | 60000 |
> >
> > *Table: A detailed comparison of model architectures, parameters, and training data.*
> >
> > **Q3: The rationale behind the LoRA rank setting is not explored.**
> >
> > **A3**: We thank the reviewer for raising this point. We present the impact of different LoRA rank values on model performance and inference speed in the table below. We evaluate different LoRA rank values on the proposed \texttt{\textit{AW}Bench}. Here, we test the inference time for 5s-clip generated video. Considering the quality of generated videos and inference speed comprehensively, we opt to set the LoRA rank to 256.
> > | LoRA rank setting | Imaging Quality | Motion Smoothness | Subject Consistency | Inference time |
> > | :--- | :--- | :--- | :--- | :--- |
> > | 64 | 4.45 | 4.28 | 4.12 | 108s |
> > | 128 | 4.53 | 4.32 | 4.26 | 110s |
> > | **256** | **4.72** | **4.56** | **4.35** | **113s** |
> > | 512 | 4.75 | 4.58 | 4.35 | 119s |
> >
> > *Table: A comparison of different LoRA rank settings based on End-to-End DreamActor-M2.*
> >
> > **Q4: When using MLLM to evaluate the synthesized pairs, which multimodal large model was used? Since large models can sometimes generate hallucinations, how can we ensure the reliability and semantic coherence of the generated pairs? What is the accuracy of the filtering process?**
> >
> > ***A4*: The proposed evaluation and validation scheme is rigorously designed with sufficient evidence, effectively demonstrating the reliability of Gemini2.5 Pro in assessing the video quality generated by video-driven models and filtering data. In Appendix H, we randomly selected 5 reference images and 3 driving videos, which were processed by Gemini 2.5 Pro to generate descriptive texts. Subsequent manual scoring of these texts revealed that all generated content achieved the highest manual scores, verifying the reliability of Gemini2.5 Pro in understanding appearance information and motion semantic information. To further explore Gemini2.5 Pro’s capability in filtering synthesized data, we randomly selected 200 synthesized data triples (reference image, driving video, generated video) and performed both automated scoring via Gemini2.5 Pro and manual scoring. A scenario was marked as correctly filtered when the scores from Gemini2.5 Pro matched the manual scores, resulting in a final filtering accuracy of 94.8\%. In Appendix I, we further demonstrate Gemini2.5 Pro’s proficiency in understanding the appearance and motion semantics of synthesized data triples. By comparing the appearance descriptions of reference images and generated images, we found the descriptions to be largely consistent, with perfect alignment between Gemini2.5 Pro’s automated scores and manual scores. Similarly, the motion semantic descriptions of driving videos and generated videos were essentially consistent, with consistent scores from both automated and manual evaluations. These quantitative filtering accuracy results and qualitative visualization results collectively confirm the reliability of using Gemini2.5 Pro for synthesized data filtering.
> >
> > **Q5: Does the model’s temporal consistency and visual fidelity degrade when generating longer video sequences?**
> >
> > **A5**: We presented the quantitative evaluation results of long video generation in R1 A1 above. Additionally, in Appendix F, we provided visualization samples of 3 long videos, from which it can be observed that our method outperforms other SOTA methods in long video generation while maintaining excellent subject consistency and motion consistency. Compared with the quantitative metrics of the original 5-second clips, the metrics for long videos have decreased, which is attributed to the lack of dedicated training for long-duration scenarios. Nevertheless, our method achieves the best performance compared with other existing driving methods.
> >
> > **Q6: Is there a specific justification for setting the LoRA rank to 256?**
> >
> > **A6**:  We have provided a detailed response in \textbf{A3} above. When we set the LoRA rank to 256, we can achieve a relatively optimal balance between the quality of generated videos and inference time.

---

### Note · Authors · 2026-01-23

I have read and agree with the venue's withdrawal policy on behalf of myself and my co-authors.